# Latent endogenous giant viruses drive active infection and inheritance in a multicellular algal host

Carole Duchêne [1], Rory J. Craig [1,2], Claudia Martinho [1,3], Rémy Luthringer[1], Ferran Agullo[1], Katharina Hipp[1], Pedro Escudeiro [1], Vikram Alva [1], Fabian B. Haas [1] & Susana M. Coelho [1]✉

Endogenous viral elements inserted in host genomes are often regarded as inert relics of past infections. Whether they can retain infective potential and contribute to active viral cycles has remained largely unresolved. Here we demonstrate that giant viral elements in the multicellular alga *Ectocarpus* can reactivate and drive productive viral infections. Using long-read sequencing and transcriptomics, we identify full-length, transcriptionally active phaeoviruses integrated within the host genome, and we use classical genetics and CRISPR–Cas to demonstrate that these elements are stably inherited through the germline, while their reactivation is precisely regulated by developmental and environmental cues including temperature. We resolve the genomic integration sites and propose a mechanism for phaeovirus integration and replication. Our work provides direct evidence and uncovers the mechanisms by which giant viral elements can reactivate, replicate and transmit both horizontally and vertically in a multicellular eukaryote, establishing a new model of latency, inheritance and evolutionary impact of giant dsDNA viruses.

Viruses are the most abundant biological entities in the ocean, reaching concentrations of millions to hundreds of millions of particles per millilitre of seawater. They infect a wide range of marine life from bacteria to giant kelps, playing key roles in ecosystem dynamics. Over the past decade, technological breakthroughs and global-scale sampling efforts have pushed viral ecology forwards, particularly in the study of microalgae and planktonic viruses. These efforts have uncovered new viral lineages, clarified infection strategies during microalgal blooms and revealed the profound influence of viruses on oceanic biogeochemical processes[1–3].

In stark contrast to the well-studied interactions between viruses and marine microbes, our understanding of viral dynamics in marine multicellular organisms, such as macroalgae, remains limited. Brown algae, members of the Stramenopile lineage, are primary producers and foundational species in coastal ecosystems[4,5], where they play key ecological roles and are increasingly threatened by climate change[6–8]. Although virus-like particles were first observed in brown algae as early as the 1970s[9], research on macroalgal viruses has lagged behind, despite the huge ecological importance of their hosts[10,11].

Among the limited number of viral lineages known to infect macroalgae, phaeoviruses, large double-stranded DNA viruses belonging to the phylum *Nucleocytoviricota*, stand out owing to their complex life cycle. First identified in the model brown alga *Ectocarpus* sp. following the observation of virus-like particles[9], phaeoviruses have become emblematic of algal viral systems. These viruses specifically target reproductive structures, converting gametangia and sporangia into virus-producing factories, thereby halting the formation of gametes and spores[12,13]. Strikingly, viral symptoms persist across generations

[1]Max Planck Institute for Biology Tübingen, Tübingen, Germany. [2]Present address: School of BioSciences, University of Melbourne, Parkville, Victoria, Australia. [3]Present address: Division of Plant Sciences, University of Dundee's School of Life Sciences, The James Hutton Institute, Dundee, UK. ✉e-mail: susana.coelho@tuebingen.mpg.de

**Table 1 | Characteristics of the EVEs identified in *Ectocarpus* genomes**

| *EVE* | Host | Chromosome | Length (bp) | Percentage GC | Genomic context | Phylogenetic group | No. of genes (including with introns) |
|---|---|---|---|---|---|---|---|
| *EVEa* | Ec01 | chr09 | 346,000 | 51.70% | 3′ UTR of gene Ec-09_002890 | A | 348 |
| *EVEb* | Ec17 | chr06 | 406,900 | 52.00% | Intergenic | A | 476 |
| *EVEc* | Ec17 | chr16 | 362,600 | 52.50% | Intron of gene Ec-16_002740 | A | 336 (14**) |
| *EVEd* | Ec17 | chr26 | 349,400 | 51.80% | Intergenic | B | 407 |
| *EVEe* | Ec17 | chr03 | 288,200 | 51.50% | Intergenic | B | 277 |
| *EVEf* | Ec267 | chr15 | 404,000* | 51.90%* | Intron of gene Ec-15_001970 | A* | 439* |
| *EVEg* | Ec267 | chr17 | 404,000* | 51.90%* | Intron of gene Ec-17_000940 | A* | 439* |
| *EVEh* | Ec267 | chr23 | 404,000* | 51.90%* | Intron of gene Ec-23_003820 | A* | 439* |
| *EVEi* | Ec267 | chr23 | 404,000* | 51.90%* | Intron of gene Ec-23_003700 | A* | 439* |

*EVEf–h are identical except for small insertions and deletions (in particular transposable element insertions; Supplementary Table 3), but were assembled as one EVE. **EVEc was manually annotated for introns.

in laboratory-maintained lines and segregate in a Mendelian manner[14] suggesting that the viral genome may integrate into the host and be transmitted vertically[15]. This hypothesis has been reinforced by the discovery of endogenous viral elements (EVEs) in several brown algal genomes[10]. These are large insertions, sometimes spanning hundreds of kilobases, that carry hallmark genes of *Nucleocytoviricota*[10,16,17]. Only two phaeovirus genomes have been sequenced in their entirety, *Ectocarpus siliculosus* virus 1 (EsV-1)[18] and *Feldmannia* species virus 158 (FsV-158)[19]. Despite their prevalence, the functional relevance of these elements remains elusive, as none of the algal genomes harbouring complete EVEs exhibit viral symptoms or detectable transcriptional activity[10,16,17].

In this study, we employ long-read genome assemblies to explore the EVE landscape across multiple *Ectocarpus* species. Crucially, we show that several *Ectocarpus* lines harbour complete EVEs and identify a transcriptionally active element that reactivates specifically in reproductive cells, where it drives hallmark viral symptoms. Through classical and reverse genetic approaches, we demonstrate vertical transmission of this element via the germline of the host and provide direct evidence that its reactivation is regulated by both developmental and environmental cues. By resolving the genomic integration sites, we further propose a mechanism for phaeovirus integration and replication. These findings not only uncover previously uncharacterized members of the *Phaeovirus* lineage, but also reveal a viral life cycle strategy that couples genomic integration, cell-type-specific reactivation and vertical transmission. This expands current paradigms of viral latency and inheritance beyond well-studied animal and plant systems, to include complex marine algae.

## Results

### Several complete EVEs in *Ectocarpus*

To bridge the gap between genomic data and virus–host biology in brown algae, we generated high-quality, chromosome-level assemblies using long-read sequencing for three *Ectocarpus* lines: Ec01 (diploid line, previously used in foundational studies of EsV-1 (ref. 14)), Ec17 (diploid line, the parent of the reference Ec32 line containing a full-length EVE[16]) and Ec267 (diploid line, which was observed to have viral symptoms in culture) (Extended Data Fig. 1 and Supplementary Tables 1 and 2). Screening these genomes with ViralRecall[20] revealed multiple integrations of *Phaeovirus* sequences within algal chromosomes (Table 1). These EVEs ranged from 288 to 407 kb in size and were inserted into distinct loci and genomic contexts. The line Ec01 contained a single

EVE (EVEa) that was identical to the EsV-1 genome[18], with the exception of a 3.5-kb variable region accumulating more than 180 mutations (Supplementary Table 3). By contrast, the two other lines, Ec17 and Ec267, each harboured four putative full-length EVEs: EVEb–e in Ec17 and EVEf–i in Ec267 (Table 1). Gene annotation and phylogenetic analysis of core viral markers (for example, DNA polymerase and major capsid protein (MCP)) revealed that the four EVEs in Ec17 derive from distinct viral clades (EVEb and EVEc from clade A, EVEd and EVEe from clade B, following the classification in ref. 21; Fig. 1a, Table 1 and Extended Data Fig. 2). By contrast, the four EVEs in Ec267 were nearly identical across coding regions (Supplementary Table 3), suggesting either multiple integrations in a single infection event or recent waves of infection by the same virus. Together, these results show that *Ectocarpus* strains harbour highly divergent EVE repertoires, reflecting either independent infections by distinct viruses (Ec17) or recent expansion events of a single viral lineage (Ec267).

These EVEs shared 48 orthogroups with the two reference genomes of phaeoviruses, EsV-1 and FsV-158 (refs. 18,19) (Fig. 1b), including ten orthogroups representing hallmark *Nucleocytoviricota* genes (Supplementary Table 4). The first and second largest group of orthologous genes are shared within each clade of virus (Fig. 1b). A synteny analysis presented the same pattern, with a strong synteny conservation within each clade, while the position of core genes was reshuffled between clades (Fig. 1c).

Despite their similarity, the four EVEs from Ec267 can be differentiated by polymorphic insertions of a DNA transposon (Extended Data Fig. 3), which is collectively present in seven different sites across the EVEs. The transposon can be classified into the prokaryotic *IS4* family based on its transposase protein and 8-nt target site duplication (Extended Data Fig. 3). Similar transposons exist in EsV-1 (refs. 22,23) and in an EVE inserted in the alga *Porterinema fluviatile*[10,17]. Many giant virus transposons are prokaryote-like[24] and, as expected, the *IS4* element is restricted to the EVE and not present elsewhere in the algal genome. As in EsV-1, the transposon from EVEf–i carries a gene encoding a Fanzor RNA-guided endonuclease, which are frequently carried by transposons of *Nucleocytoviricota* viruses[23].

Analysis of the genomes of Ec17 and Ec267 revealed that each EVE is present in a hemizygous state, and we identified exact EVE insertion sites by comparison to the homologous, EVE-free chromosome. All EVE insertions occurred at 'CC' dinucleotides (Fig. 1d and Extended Data Fig. 4), in introns, 3′ UTRs or intergenic regions (Table 1). Delaroque et al.[18] identified a possible integrase gene in EsV-1 encoding

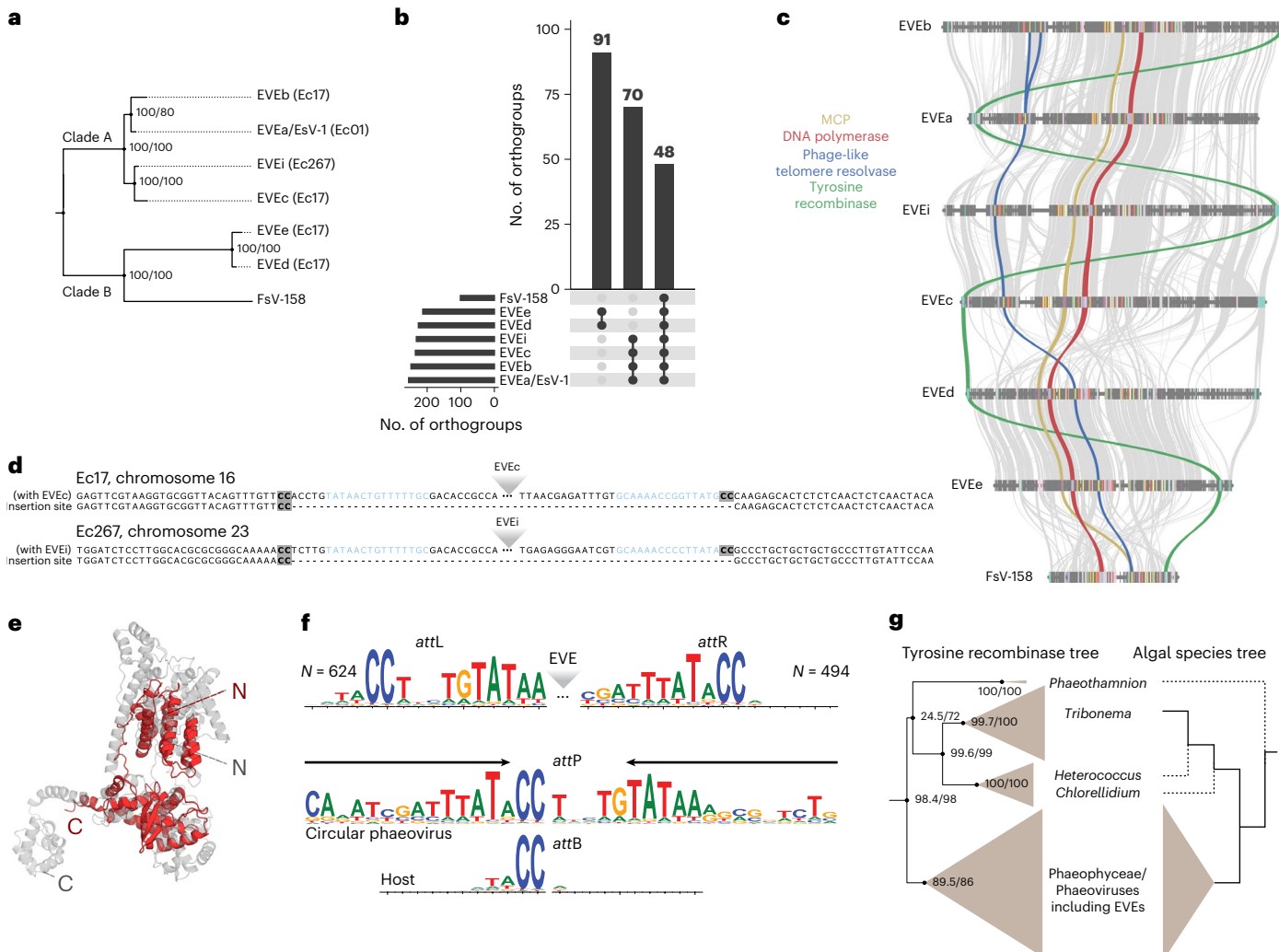

**Fig. 1 | Features of the *Ectocarpus* EVEs and their insertion sites.**
**a**, A phylogenetic tree of the EVEs and reference viruses (EsV-1 and FsV-158) inferred from 40 shared single-copy genes. See also Extended Data Fig. 2. The *Ectocarpus* lines in which these EVEs are present are indicated in parenthesis.
**b**, Groups of orthologous genes shared among EVEs and phaeoviruses (the three biggest groups are shown). **c**, Synteny of the EVEs based on the shared orthologous genes. The key genes mentioned in the text are highlighted.
**d**, Example insertion sites of EVEc and EVEi. The 'CC' dinucleotide is highlighted in bold with a grey background and the inverted repeats are in blue. **e**, Predicted structure of the putative integrase (tyrosine recombinase of EVEc) in grey;

aligned to a XerH recombinase monomer from *Helicobacter pylori* (PDB: 5JJV, chain A) in red. Root mean square deviation of 4.81 Å, template modelling score ≈0.69. See also Extended Data Fig. 5. **f**, Sequence logo for the left end/*att*L and right end/*att*R termini and flanking sequences identified in Phaeophyceae genomes, and the proposed mechanism for integration with homology between host (*att*B) and viral (*att*P) attachment sites. The arrows indicate *att*P inverted repeat. **g**, Left: a phylogenetic tree of the tyrosine recombinase proteins found in Phaeophyceae and in closely related algae. Right: a schematic algal phylogenetic tree based on refs. 85,86. Uncertain taxa positions are marked with dashed branches.

a tyrosine recombinase, which mediate the genomic integration of several prophages and integrative conjugative elements in prokaryotes[25,26] and eukaryotic mobile elements[27,28]. The putative tyrosine recombinase was also found in FsV-158 (ref. 19) and is present in all identified EVEs, constituting one of the 48 universal orthogroups (Fig. 1b). Notably, the gene is generally located near either the left or right end of the EVEs (Fig. 1c). Similar organizations are common in phage genomes and constitute integration cassettes[26,29]. The phaeoviral tyrosine recombinase structure modelled with AlphaFold is consistent with an active enzyme, with the conservation of key catalytic residues (Fig. 1e and Extended Data Fig. 5), supporting that the candidate phaeoviral integrases are functional tyrosine recombinases.

Integration via tyrosine recombinases typically involves recombination between attachment sites present in the host genome and the circular integrative element (*att*B and *att*P, respectively, following phage nomenclature)[30]. Once integrated, the host–virus junctions

constitute new attachment sites, *att*L and *att*R, and recombination between these sites results in virus excision. Circular maps have been reported for EsV-1 and FsV-158 (ref. 18) (see also below), consistent with such a mechanism in phaeoviruses. The extent of sequence homology between the attachment sites varies extensively[31] but is relatively simple in eukaryotic mobile elements, enabling integration at multiple sites throughout the genome[32]. In addition to the 'CC' insertion site, all EVEs feature a corresponding 'CC' dinucleotide (arbitrarily drawn as the left end terminus in Fig. 1d), consistent with the tyrosine recombinase acting on minimal homology of two nucleotides between the viral *att*P and algal *att*B sites.

The left and right terminal sequences (which combine to form *att*P in the circular form) are conserved among the EVEs, potentially representing binding motifs for the tyrosine recombinases. The left end sequence features an inverted repeat that is conserved among the EVEs (Extended Data Fig. 6a, red), while the right ends

feature imperfect direct repeats that are more variable among EVEs (Extended Data Fig. 6b). *att*P sites frequently feature inverted repeats flanking the core sequence that undergoes recombination, for example, in Lambda phage[33]. Indeed, we identified a conserved AT-rich inverted repeats flanking the core 'CC' dinucleotide in the EVE *att*P sites (Fig. 1d, blue, and Extended Data Fig. 6a). Intriguingly, the ends of EVEe are substantially divergent from the remaining EVEs (especially the right end), despite EVEe and EVEd exhibiting a close phylogenetic relationship (Fig. 1a).

We next screened other brown algae genomes for viral insertions using two hidden Markov models (HMMs) built from alignments of the conserved EVE left and right ends (Extended Data Fig. 7). Owing to assembly fragmentation, we were only able to recover complete EVEs (that is, left and right ends located on the same contig) in four genomes, all of which showed CC–CC host–virus junctions (Extended Data Fig. 7d). In addition, we retrieved hundreds of putative left and right end sequences from standalone contigs, from which we generated consensus logos of the attachment sites (Fig. 1f). These revealed that *att*P-*att*B homology may extend over four to five nucleotides, with a weak contribution from the first three bases (consensus 'ATA'), followed by the nearly invariant CC dinucleotide.

Taken together, our results support the idea that circular phaeoviruses integrate via a conserved tyrosine recombinase requiring minimal sequence homology. However, the diversity of attachment sites appears greater than previously appreciated. Notably, we found no homology between the EVE attP and FsV-158, whose insertions feature GC–GC host–virus junctions rather than CC–CC[34]. Interestingly, Meints et al.[34] reported an AT-rich inverted repeat adjacent to the core 'GC' dinucleotide of FsV-158, suggesting this may be a common feature of phaeovirus *att*P sites. Thus, our findings are consistent with a conserved insertion mechanism among phaeoviruses, with at least two categories of host–virus junctions evolving in the group.

Finally, as previous work reported the presence of viral genes in an algal group basal to Phaeophyceae, the Xanthophyceae (and proposed the clade xanthovirus as sister clade to phaeoviruses), we searched for the presence of phaeovirus-like tyrosine recombinases in those algae. We found homologues to the phaeovirus tyrosine recombinases in the genomes *Tribonema*, *Hetereococcus* and *Chlorelidium* (Xanthophyceae), suggesting that xanthoviruses and related viruses may also integrate into their host's genome (Fig. 1g and Supplementary Table 5). Remarkably, the congruence of the algae clade phylogenies and the tyrosine recombinase protein phylogeny strongly suggests co-evolution of the viruses and their hosts, potentially suggesting the maintenance of an integrative lifestyle in these giant viruses for more than 400 million years[10].

### Vertical transmission of EVEs leads to symptomatic algal progeny

To link the presence of EVEs with viral symptoms, we generated segregating progenies from the two diploid individuals, Ec17 and Ec267 (Fig. 2a). Two hundred haploid individuals originating from independent meiosis from Ec17 were grown at 10 °C, a condition known to induce viral symptoms in other *Ectocarpus* species[12], and screened for infection symptoms by microscopy and by qPCR targeting the MCP gene (Fig. 2b and Extended Data Fig. 8). Of the haploid progeny, 47.5% showed symptoms, which, given the hypothesis that the EVEs are causing the symptoms, suggests that only one of the four EVEs was associated with active infection (Supplementary Table 6). Indeed, genotyping for the EVE content of each individual revealed full linkage between symptoms and presence of EVEc (Fig. 2c, $\chi^2$ test $P = 2.64 \times 10^{-35}$), whereas the other EVEs were not linked with symptoms ($\chi^2$ test $P > 0.05$). The EVE insertions segregated in a Mendelian fashion as the different EVE combinations were obtained in equal proportions and not linked to a specific sex (Supplementary Table 6). We found no change in the severity of the symptoms associated with the EVE

combinations, suggesting lack of interaction between the different EVEs (Extended Data Fig. 8).

We generated 140 meiotic progenies from the Ec267 strain (Fig. 2a). Intriguingly, and in contrast to the Ec17 segregating population, 90.7% of the Ec267 progeny displayed symptoms (Supplementary Table 6). Genotyping revealed that all individuals inherited at least one EVE. Although EVEh and EVEi have previously been assembled on the same contig, analysis of the segregating progeny showed that each individual inherited either EVEi or EVEh, alone or in combination with EVEf and EVEg (Fig. 2d). These results indicate that both EVEi and EVEh reside on different homologues of chromosome 23. Moreover, they demonstrate that both EVEi and EVEh represent functional proviruses capable of inducing symptoms.

To assess vertical transmission of EVEs and associated symptoms through syngamy, we crossed a haploid male from the Ec17 progeny carrying EVEb (a putatively inactive element, see above), with a female from the Ec267 progeny carrying EVEi (Fig. 2a). Among the haploid progeny from the resulting diploid individual (Ec704), 41.4% displayed viral symptoms consistent with Mendelian segregation of symptoms ($\chi^2$ test, $P = 0.0713$). Strikingly, all symptomatic individuals carried EVEi (Fig. 2f; $\chi^2$ test, $P = 3.34 \times 10^{-17}$), whereas EVEb again showed no association with symptoms (Supplementary Table 6). Thus, across all examined algae, the occurrence of viral symptoms is strictly and exclusively linked to the presence of an active EVE in the genome.

Finally, to unambiguously demonstrate that functional EVEs drive the observed viral symptoms, we used CRISPR–Cas to excise EVEc from the genome of a haploid individual carrying this element. Guide RNAs were designed against host sequences flanking the EVEc integration site, yielding two independent crispants lacking the full insertion (Fig. 2f). Phenotyping of these mutants lacking EVEc reconstituted a healthy alga, devoid of viral symptoms (Fig. 2g,h), thereby establishing a direct causal link between active EVEs and symptom expression.

### EVEc latency and activation

We found that viral symptoms manifest only during reproductive stages and under defined environmental conditions. Temperature appears to play a critical role in triggering viral activation: strong symptoms were observed when algae were grown at 10 °C, whereas the same algal strain remained asymptomatic at 20 °C (Fig. 2b). To investigate the mechanisms underlying EVEc activation, we performed RNA-sequencing (RNAseq) on haploid algae under symptom-inducing and non-inducing conditions. This analysis revealed robust transcriptional activation at the EVEc locus at 10 °C, coinciding with the appearance of symptoms (Fig. 3a). By contrast, the locus remained largely transcriptionally silent at 20 °C, except for a small subset of 18 genes that were constitutively expressed (Extended Data Fig. 9).

Interestingly, a small fraction of EVEc genes (14/372) contain introns, with splice sites consistent with those reported for *Ectocarpus*[35] (Extended Data Fig. 10). Introns were also detected in core Phaeoviral genes, including putatively essential genes such as DNA polymerase, the MCP and an Apoptosis inhibitory protein 5-like protein (EVEc-000140). Given that EVEc is clearly active and induces symptoms (Fig. 2a,c), the presence of introns does not indicate viral decay but may instead be important for correct mRNA processing by the host cell machinery[36]. Spliceosomal introns have also been detected in other *Nucleocytoviricota*, in particular in chlorovirus core genes[37,38].

It has previously been noted that both EsV-1 and FsV-158 encode a telomere resolvase (also known as protelomerase)[19,39], and we recovered the corresponding gene as universally conserved among the *Ectocarpus* EVEs (Fig. 1c). Such an enzyme is employed by the phage N15 to transition from a circular to a linear genome with covalently closed ends[40] and it has been hypothesized that phaeoviruses could similarly exist in both circular and linear forms[19,39]. To investigate this, we performed Oxford Nanopore gDNA sequencing of symptomatic algae, revealing

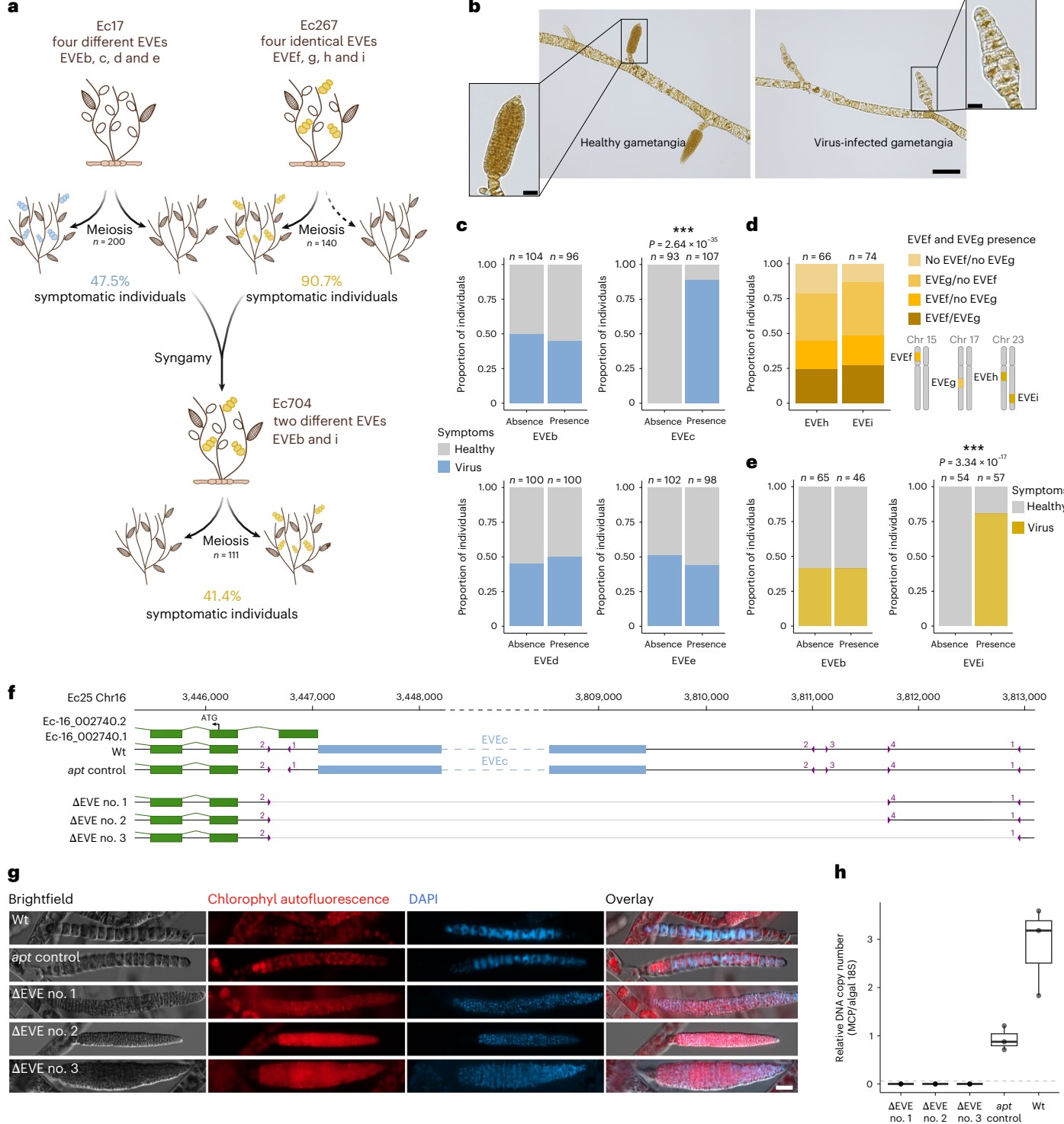

**Fig. 2 | Viral symptoms are caused by vertical inheritance of EVEs.**
**a**, Segregation of symptoms in Ec17, Ec267 and Ec704 progenies.
**b**, Asymptomatic and symptomatic reproductive organs in Ec17 offsprings. Scale bars, 100 μm, 20 μm (inset). **c**, The association between EVEs and symptoms in Ec17 progenies, tested with a $\chi^2$ test. **d**, Segregation of EVEs in Ec267 progenies. Note that 90.7% of the individuals are symptomatic. **e**, The association between EVEs and symptoms in Ec704 progenies, tested with a $\chi^2$ test. **f**, A scheme of the genetic modifications upon CRISPR–Cas transformation at the EVEc locus. The guide RNA binding sites are depicted in purple with numbers indicating the guide name on each EVE flank (see also Supplementary Table 8) and the grey lines indicate deletions. The numbers in black at the top indicate the nucleotide

position on chromosome 16. **g**,**h**, Assessment of symptoms in the EVEc deletion mutants by microscopy on the gametangia (**g**) and qPCR (**h**). Note the swollen cells and the DAPI staining distributed throughout the entire cell in infected organs, in contrast to the discrete, dot-like signal observed in gametes within healthy gametangia. In **h**, the box plot represents the median of three biological replicates, with the minima and maxima of the box corresponding to the first and third quartiles, respectively, and whiskers to 1.5 times the interquartile range. Wild type (Wt), haploid individual from Ec17 progeny (strain Ec25, Supplementary Table 1), carrying EVEc; *apt* control, control line mutated at the *APT* locus (that allows for selection of mutants) and in the EVE left flank without the deletion of EVEc. ***$P < 0.001$.

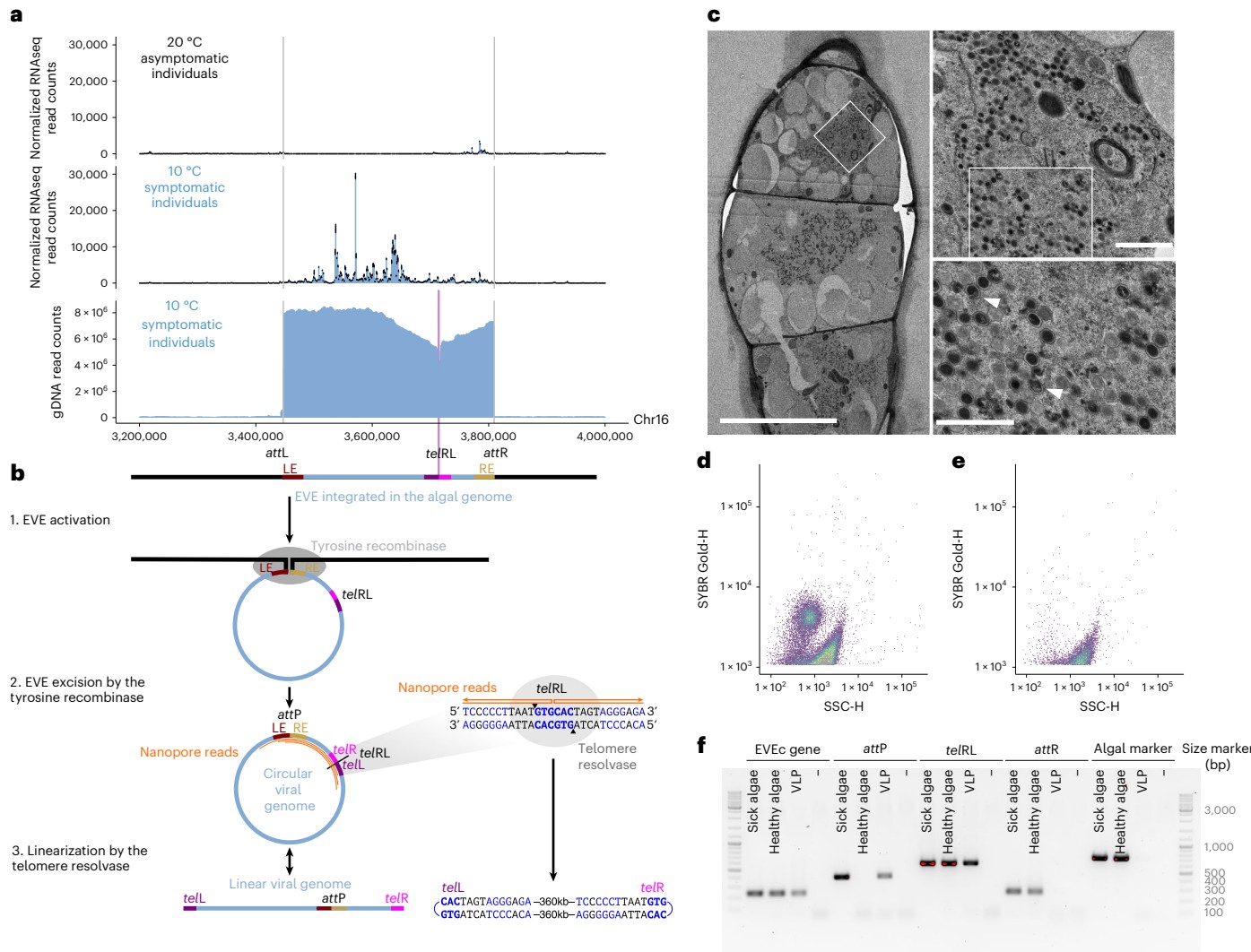

**Fig. 3 | Activation of EVEc and formation of virions at low temperature.**
**a**, Genomic activity at the EVEc locus in gametophytes grown at 10 °C and 20 °C. RNAseq expression across the EVEc locus for algae grown at 20 °C (asymptomatic; top) or at 10 °C (symptomatic, middle) and genomic long-reads density at 10 °C (symptomatic, bottom). Vertical grey lines mark the EVEc boundaries. **b**, The proposed mechanism for EVEc excision supported by Nanopore read mapping, schematized in orange. LE and RE denote the left and right ends of the EVE insertion, respectively, which recombine to form the *att*P site. Blue nucleotides in *tel*RL site are consistent with an inverted repeat and the black triangles represent putative cleavage sites 6-bp apart at the core palindromic site. **c**, Transmission electron microscopy of symptomatic cells.

The cellular content is occupied by vacuoles and virus particles (left). Zoom on the virus particles (right). The arrowheads indicate the putatively immature virus particles. Scale bars, 10 μm (left), 1,000 nm (top right), 700 nm (bottom right). Representative sick gametangia out of four imaged gametangia. **d,e**, Flow cytometry analysis of particles released from an alga with viral symptoms (**d**), showing a virion-like population, or from a healthy alga (**e**). **f**, PCR on extract from sick algae, grown at 10 °C (first lane), healthy algae grown at 20 °C (second lane) or on DNase-treated virion-like particles (VLP, third lane), to amplify an EVEc gene marker (GP1 gene, see Supplementary Table 7 for all the primers), the *att*P site, the *tel*RL site, the *att*R site or an algal marker (ITS1). Viral release, flow cytometry and PCR were performed twice independently.

high read coverage across the EVEc locus that supports active replication of viral DNA (Fig. 3a). We observed reads spanning the viral *att*P site, consistent with circularization following recombination between the integrated *att*L and *att*R sites (Fig. 3a,b). However, we also noticed a drop in gDNA read coverage at an internal position of EVEc (Fig. 3a). In phage N15, telomere resolvase acts on a 56-bp inverted repeat called the *tel*RL site, generating a cleavage with a 6-nt stagger at the core of the palindrome that is subsequently resealed as the covalently closed telomeres *tel*L and *tel*R[40]. Inspection of the reads mapping to the internal breakpoint in EVEc revealed that these reads map in a sense orientation for part of their length, before they revert and map to the corresponding region in an antisense orientation. These reads exist both upstream and downstream of the EVEc breakpoint, and remarkably they all revert at an exact site that is centred on a 6-bp palindrome present within a larger inverted repeat (Fig. 3b). Since genomic DNA

becomes single stranded during Nanopore sequencing, it is possible that these sense–antisense mapping reads correspond to covalently closed telomeres (Fig. 3b), although they may also be derived from replication intermediates ('Discussion').

Overall, we propose that the viral tyrosine recombinase mediates EVE excision and circularization via recombination (Fig. 3b). Reads mapping across *att*P and a putative linearization site support a circular viral genome, consistent with data supporting circular maps for EsV-1 and FsV-158 (refs. 18,19). However, the genome probably undergoes linearization at a specific palindromic site that strikingly resembles the *tel*RL site of phage N15. A conserved viral telomere resolvase, homologous to that of phage N15 (ref. 19), may cleave this *tel*RL site and reseal the breaks to form covalently closed left (*tel*L) and right (*tel*R) telomeres (Fig. 3b). Viral replication presumably occurs in the linear form, as is the case in phage N15 (ref. 40).

Following replication and gene expression, the next stage of the viral life cycle is virion assembly within host cells. Transmission electron microscopy revealed that symptomatic cells identified by transmitted light and fluorescence microscopy (abnormal gametangia heavily stained with DAPI; Fig. 2b,g) are filled with virus-like particles resembling EsV-1 (Fig. 3c). These particles exhibited a characteristic hexagonal morphology, averaging 150 ± 6 nm in diameter, with a multilayered shell surrounding an electron-dense core (Fig. 3c). We also observed putative assembly intermediates with incompletely filled cores (Fig. 3c, arrowhead). Consistently, virus-like particles positive for the EVEc gene marker were detected by flow cytometry following induced release from symptomatic algae (Fig. 3d–f). To determine the form of the viral genome packaged within these particles, we performed PCR analyses and successfully amplified both the *att*P and *tel*RL sites. This suggests that the viral genome may be packaged in its circular form (Fig. 3f).

Together, our findings demonstrate that temperature shifts can trigger the transition from latency to productive infection specifically in gametangia, culminating in giant viral gene expression, replication, particle assembly and virion release.

## Discussion

This study reveals a previously uncharacterized latent life cycle of an integrated dsDNA (giant) virus in a multicellular eukaryote, and demonstrates its stable vertical transmission across multiple generations. In all three generations examined, viral symptoms were strictly correlated with the presence of integrated, transcriptionally active phaeoviruses. Whereas earlier work reported Mendelian segregation of symptoms, our findings provide the first direct genetic evidence linking symptom inheritance to the transmission of specific viral insertions within the algal genome, an association that had remained unresolved for decades. We further demonstrate that the virus is inherited as an integrated genomic element, rather than as an episome tethered to host chromosomes as in herpesviruses[41], underscoring a unique strategy of viral persistence in these algal hosts.

Giant EVEs, whether functional or defective, segregate in a strictly Mendelian fashion, suggesting that even large ~400-kb insertions impose no detectable fitness cost to their algal hosts. Together, these findings establish a direct genetic basis for the vertical transmission of integrated dsDNA viruses in eukaryotes and highlight a striking tolerance of brown algal genomes to massive viral integrations.

We show that EVEs in brown algae behave as proviruses: they remain transcriptionally silent for most of the life cycle but undergo tightly regulated reactivation at specific developmental stages and under defined environmental conditions. Temperature proved to be the decisive trigger. At elevated temperatures, algae completed their entire life cycle, including reproduction, without viral activation. By contrast, low-temperature conditions induced viral reactivation, converting reproductive structures into symptomatic, virus-producing organs. This conditional, stage-specific activation is a hallmark of latent viral life cycles and stands in clear contrast to acute or chronic infections[42], underscoring the finely tuned regulation of proviral reactivation in multicellular alga hosts. A similar life cycle was recently proposed for the punuiviruses infecting the unicellular green alga *Chlamydomonas reinhardtii*. However, in that case, the triggers of viral activation remain unidentified. By contrast, our *Ectocarpus*–*Phaeovirus* pathosystem currently represents, to our knowledge, the only experimentally tractable model with clearly defined, inducible conditions for studying latent infections by members of the *Nucleocytoviricota*.

The features of phaeovirus EVE insertion sites point to an integration mechanism distinct from that of the *Chlamydomonas* punuiviruses, which feature large terminal inverted repeats and flanking target site duplications at integration sites[43]. These contrasting genomic signatures indicate that the two viral groups employ fundamentally different enzymatic strategies for integration: while punuiviruses are thought to rely on a retroviral-like DDE/D integrase (possibly of virophage origin)

our data implicate a tyrosine recombinase in phaeovirus integration. Taken together, evidence from *Ectocarpus* and *Chlamydomonas* supports independent acquisitions of integration machinery among giant viruses, highlighting convergent evolutionary trajectories toward a latent lifestyle in these two *Nucleocytoviricota* families.

Interestingly, these integration mechanisms mirror those observed among diverse prophages, with integration via a DDE/D integrase exemplified by Mu and via recombinase by Lambda[30,44]. Furthermore, as with eukaryotic mobile elements and some prokaryotic integrative conjugative elements that integrate via tyrosine recombinases[26–28,31], the homology requirements between the phaeovirus and algal host attachment sites appear to be minimal. This presumably facilitates integration across the genome, which may be especially important given the high density of intact and degrading EVEs in many genomes. We further explored the sister clade of brown algae, the *Xanthophyceae*, which includes both unicellular and multicellular taxa and harbours genomic traces of xanthoviruses, the sister group to phaeoviruses. The presence of phaeovirus-like tyrosine recombinase genes in these genomes raises the possibility that xanthoviruses share a similar integration-based, latent lifestyle. This, in turn, supports the hypothesis that latency via genomic integration may represent an ancient strategy, predating the emergence of complex multicellularity in brown algae, more than 400 million years ago[45]. Although the precise boundaries and attachment sites of these EVEs remain to be defined, these findings open compelling avenues for investigating latent viral infections across diverse algal lineages.

Our data on the active infection of EVEc advance several proposed models for the life cycle of phaeoviruses. Although data supporting linear virus genomes was previously inconsistent or missing, the presence of a conserved telomere resolvase gene in EsV-1 and FsV-158 had led to the idea that phaeovirus genomes could transition between circular and linear forms. Such transitions play multiple roles in the diverse life cycles of viruses, with examples of circularization including Lambda phage[30] and herpesviruses[46]. Phage N15 transitions from a circular to linear molecule, forming an unintegrated prophage plasmid with covalently closed ends[40]. We find a linearization site in EVEc with striking resemblance to the *tel*RL site of phage N15, and we propose that the phaeovirus telomere resolvase act on these sites to produce linear molecules with covalently closed telomeres. In phage N15 and the covalently closed linear chromosomes of the Lyme disease pathogen *Borrelia burgdorferi*, telomere resolvase is further involved in DNA replication[40,47]. Replication initiates from a single origin of replication and, at least in *B. burgdorferi*, this results in a circular dimer replication intermediate that is resolved by DNA breakage and resealing of the telomere–telomere junctions by the telomere resolvase[48]. Schroeder et al.[19] identified a possible internal origin of replication in FsV-158, suggesting that the *Borrelia* model could extend to phaeoviruses. However, we note that all of the analysed EVEs in this study also encode an A22-like Holliday junction resolvase, which is conserved in ~75% of *Nucleocytoviricota* genomes[49]. In linear vaccinia virus (poxviruses), replication is initiated at the telomeres and the A22-Holliday junction resolvase cleaves concatemer replication intermediates[50,51]. Together, our observations highlight multiple plausible mechanisms for phaeovirus replication and open the way to targeted experiments that can now directly test how these enigmatic viruses propagate.

## Methods

### Algae cultures

Supplementary Table 1 describes the algal strains used. Algae (gametophyte or sporophyte generations) were cultured in natural seawater (NSW) or artificial seawater (ASW, classic sea salts, Tropic Marin, 40 g l⁻¹) with half-strength Provasoli solution ($0.5PE^{52,53}$) under a 12:12 light:dark cycle (~30 µmol photons m⁻² s⁻¹) at 14 °C, or under 16:8 light:dark cycle at 10 °C or 20 °C. All manipulations were performed under a laminar flow hood.

## Long-read genomic DNA sequencing

For DNA extraction and genome sequencing of Ec01, Ec267 and Ec17, algae were grown at 14 °C in NSW + 0.5PE. Before harvesting, algae were treated with antibiotic in NSW + 0.5PE for 3 days as follows: penicillin 2 g l⁻¹, streptomycin 0.5 g l⁻¹ and dhloramphenicol 0.1 g l⁻¹ (day 1); kanamycin 1 g l⁻¹ (day 2); cefotaxine 2 g l⁻¹ (day 3)[52,54]. In between each treatment, algae were rinsed with 0.5 l of sterile NSW. After the antibiotic treatment, algae were harvested on a 100-µm mesh, quickly dried on absorbing paper, weighted, frozen in liquid nitrogen and stored at −70 °C.

High-molecular-weight DNA was extracted from ~0.2 g algae as follows: frozen algae sample were ground to a fine powder in liquid nitrogen for ~15 min and resuspended in 10 ml CF lysis buffer (Macherey–Nagel). The suspension was gently mixed by inverting the tube. After 5 min, when visible lysis was observed, the following components were added: 200 µl EDTA 0.5 M pH 8, 200 µl Triton 100%, 4 ml CTAB 10%, 5 ml NaCl 5 M, 200 µl RNaseA and 400 µl Proteinase K (provided with the Macherey–Nagel HMW DNA Nucleobond kit). After gently mixing, the sample was incubated overnight at 37 °C with gentle agitation at 100 rpm. On the next day, 100 µl RNaseA and 200 µl Proteinase K were added again and the sample incubated for an extra 1.5 h. DNA extraction was continued using the HMW DNA Nucleobond kit (Macherey–Nagel) following the manufacturer's instructions and resuspended in 60 µl nuclease-free H₂O in a DNA LoBind tube. DNA quality was assessed with a Femto Pulse system (Aligent) for DNA size profile, on a Nanodrop (Ependorf) for purity and quantified with the Qubit High Sensitivity system (Invitrogen).

Long-read Nanopore library preparation was performed using the SQK-LSK114 Ligation sequencing V14 kit (Oxford Nanopore) following the manufacturer's instructions. Libraries were loaded on a PromethION Flow cell R10 in a PromethION 2 Solo (Oxford Nanopore). The flow cell was washed for 3 h every ~24 h with the EXP-WSH004 Flow cell washing kit and the same library was reloaded after wash. For each algal genome, two to three such runs were required.

## Genome assembly and EVE identification in *Ectocarpus* genomes

Basecalling was performed by Oxford Nanopore's Dorado, v0.6.2 (ref. 55) with the model dna_r10.4.1_e8.2_400bps_sup@v4.3.0. ONT long reads draft assemblies of Ec01, Ec17 and Ec267 were performed by canu v2.2[56]. Besides the default settings, the following settings were used: minInputCoverage=5 stopOnLowCoverage=5 genomeSize=440 m minReadLength=800. The assembly quality was measured by comparing the draft assemblies to the published *Ectocarpus* Ec32 reference genome[57].

To identify viral elements in the contigs of the draft assemblies, we used Viralrecall v2.1[20] in default mode. The draft assemblies were mapped to the Ec32 reference genome using minimap2 v2.28[58] to identify chromosomes. By extracting the EVE–host boundaries we obtained the contigs belonging to the homologous chromosome (without the EVE insertion), which we used to precisely characterize the EVE insertion sites.

To detect variations between closely related EVEs, Medaka v2.0.1[59] was used to perform a haploid variant calling on ONT long reads. The originally published EsV-1 genome[18] was compared with EVEa mapped reads and all EVEs located on Ec267 were compared with its reference EVEi. Medaka_variant was used with the model r1041_e82_400bps_sup_variant_v4.1.0.

## EVE gene prediction

To predict protein-coding genes in the EVE genomes, we ran Prodigal[60] (v.2.6.3) in metagenomic mode (-p meta); Prodigal-gv[61] (v.2.11.0-gv) also in metagenomic mode; and GeneMarkS-2[62] (v.1.14_1.25_lic) with the '--genome-type auto' option set. To combine and dereplicate the genes predicted by these three tools, we ran GFF-Adder from ORForise[63]

(v.1.4.2), using the GFF from Prodigal as reference and adding the GFFs from Prodigal-gv and GeneMarkS-2.

In addition to the approaches detecting mono-exonic ORF, RNAseq data (see 'RNAseq and analysis') was used to extract spliced gene models. The RNAseq mapping files were loaded to the genome browser JBrowse2. Exons were detected by manually observing the intron–exon boarders of each gene model. In total, 14 spliced gene models were found.

## Phylogeny of the EVEs

Orthofinder[64] (v2.5.5) was ran on predicted EVE proteins and FsV-158 proteome. Synteny analysis was built on the Orthofinder results and represented using MCscan (jcvi package)[65]. Phylogeny of the EVEs was built on 40 genes present in single copy in each EVE and viral genome. Proteins were aligned with MAFFT v7.526[66] (E-INS-I option), trimmed with trimAl[67] (v1.2rev59, --automated1) and the alignments concatenated. The species tree was reconstructed with IQ-TREEe[68] (v1.6.12) with partition analysis option, automated model finding, ultrafast bootstraps and SH-like approximate likelihood ratio test. The tree was rooted based on previous data[10,69] and MCP and PolB DNA polymerase trees. MCP and PolB polymerase trees were built as follows: MCP and PolB proteins were identified by running hmmsearch (HMMER V3.4, hmmer.org) on Phaeophyceae proteomes using the GVOG HMM models from[49] as queries, with an e-value cut off of e-60 for MCP and e-100 for PolB. Sequences were aligned with MAFFT[66], trimmed with trimAl[67] (v1.2rev59, -automated1) then manually curated and the phylogenetic tree reconstructed with IQtree[68] with automatic best model finding and ultrafast bootstraps and SH-like approximate likelihood ratio test (-bb 1000 -alrt 1000). The tree was rooted with *Tribonema minus* sequences.

## EVE termini in Phaeoexplorer datasets

Alignments were produced for the *Ectocarpus* EVE left and right ends separately using MAFFT v7.525[66] with the option 'L-INS-i'. The alignments were manually trimmed to the well-conserved terminal regions (see Extended Data Fig. 6a) and a profile HMM was produced for each using the HMMER v3.1b2 (hmmer.org) tool hmmbuild. All Phaeoexplorer[10] genome assemblies were then queried using the resulting profiles and the nhmmer command run with default parameters. The genomic sequence of hits of at least 80 bp that intersected with the model within 50 bp of the terminal end were then extracted and extended to include flanking sequence (upstream for the left end, downstream for the right end). Putative EVE termini with identical flanking sequences, which may represent common EVEs shared between closely related strains/species, were reduced to a single copy. Independent sequence logos for the inferred *att*L and *att*R sites (that is, putative EVEs and flanking sequence) were produced using WebLogo3[70]. The putative *att*P and *att*B sites were then inferred based on the known host–virus junctions of the *Ectocarpus* EVEs.

## Structural modelling and alignment of the tyrosine recombinase from EVEc

We noticed that the predicted protein sequence of the tyrosine recombinase from EVEc (EVEc-000150) had a N-terminal region (residues 1–68) that was not present in the tyrosine recombinases from the remainder EVEs, nor that of FsV-158 (Extended Data Fig. 5). Moreover, we speculated that the residue composition of this N-terminal segment might contribute to a disordered region. Taking into account both of these considerations, we posited that this segment was the product of a spurious start codon, inferred during the gene-calling step, and that the true start site corresponded to M69. Thus, we built a structural model of the tyrosine recombinase from EVEc lacking this N-terminal region (residues 69–860), using an installation of AlphaFold2[71] (v.2.3.1), at the Max Planck Computing and Data Facility. The prediction was carried out using the default settings. We then aligned the model with the highest confidence (ranked_0.pdb), with a monomer of the XerH

recombinase from *Helicobacter pylori* (PDB: 5JJV, chain A), using the US-align web server[72,73] with default settings. Structure visualizations were created in PyMOL v.2.5.0.

### Tyrosine recombinase in other algae

An HMM model was built using the tyrosine recombinase genes of EVEa, b, c, d and f and searched for in the Phaeoexplorer database with e-value cut off of e-10. To limit the number of truncated sequences, only hits longer than 400 amino acids were retained.

For Xanthophyceae and other algae outside Phaeophyceae, protein predictions are often not available in publicly accessible genomes and the tyrosine recombinases were searched using tblastn, with EVE tyrosine recombinases as query, against genomes from Xanthophyceae and closely related algae (Supplementary Table 5 for the list of genomes examined). Contigs containing tyrosine recombinase hits were extracted and protein sequences predicted using Prodigal[60] (v2.6.3) with default parameters. As above, an HMM search with the EVE tyrosine recombinases HMM model was performed on predicted proteins with an e-value cut off of e-10. Sequences from Phaeoexplorer and Xanthophyceae were clustered at 90% identity using CD-HIT[74]. Clustered sequences of tyrosine recombinases from phaeoviruses, xanthoviruses and bacteria were aligned using clustalO, trimmed with trimAl (-automated1). A phylogenetic tree was reconstructed with IQ-TREE with automatic model finding, ultrafast bootstraps and SH-like approximate likelihood ratio test (-bb 1000 -alrt 1000).

The genomes of *Schizocladia* and *Chrysoparadoxa* contained only fragments of tyrosine recombinase genes. To assess their phylogenetic position, these fragments (longer than 100 amino acids) were aligned with the algal proteins mentioned above using hmmalign and IQ-TREE was run with the best model parameters found above.

### Assessment of viral symptoms in segregating progenies by microscopy and qPCR

Segregating progenies from the different sporophytes were generated as follows: *Ectocarpus* sporophytes were grown at 14 °C in NSW + 0.5PE in high light (50 µm m$^{-2}$ s$^{-1}$) for 4 weeks to induce unilocular sporangia production (see also Extended Data Fig. 1 for *Ectocarpus* life cycle; only one meiosis occurs per uniclouar sporangia, followed by several rounds of mitosis). Individual unilocular sporangia were isolated by microdissection into small Petri dishes as explained in ref. 53, where they released meiospores. Germlings were grown at 14 °C in NSW + 0.5PE in low light for 3 weeks. After 3 weeks, one gametophyte per unilocular sporangia was isolated and grown at 10 °C for 5 weeks in ASW with 0.5PE. Each individual gametophyte was sampled for fluorescence microscopy, qPCR and genotyping. For fluorescence microscopy, material was mounted between slide and coverslip in Vectashield mounting media with DAPI (Vector laboratories) diluted by half in seawater. The material was imaged under a Zeiss Axiovert Observer Z1 microscope in brightfield, DAPI (excitation LED 385/30 nm, emission filter 405–450 nm) and chlorophyll autofluorescence (excitation LED 631/33 nm, emission filter 653–780 nm) channels.

For qPCR and genotyping, samples were ground at room temperature with a 1-mm metal bead for twoce 1 min at 30 Hz with a Tissue Lyser II bead beater (Qiagen) in 50 µl water. The sample was then diluted 40 to 100 times in water depending on the gametophyte fragment size, and stored at −20 °C.

qPCR was performed in technical triplicates on the ground algae with the Sso Advanced Universal inhibitor-tolerant SYBR Green supermix (BioRad) with 4 µl of sample in a 10 µl reaction. Primers were designed to amplify the MCP gene for EVEa, b, c and f–i, and primers from ref. 75 were used to amplify the algal 18S (Supplementary Table 7). qPCR was run on a CFX96 BioRad real-time PCR system with 5 min 98 °C denaturation followed by 40 cycles of amplification (15 s at 98 °C −30 s at 62 °C) and a final melting curve. Relative copy number were calculated using $\Delta C_q$.

### Genotyping PCR

Primers (Supplementary Table 7) were designed to amplify either EVE genes (with more than 4 SNPs between different EVEs, used in particular to distinguish EVEd and EVEe) or EVE–host boundaries (EVEb and c, EVEf, g, h and i). PCR was performed on the same ground algal sample as for the qPCR (see above) with the Terra direct PCR kit (Takara) (see Supplementary Table 7 for the amplification cycle programs). Segregation analysis and EVE-symptoms associations were done with Chi-square tests in R (v4.3.0).

### Generation of EVE deletion by CRISPR–Cas

For the in-depth study of viral symptoms caused by EVEc, we focused on the algal strain Ec25 (female, with EVEc and EVEd), which is the model female gametophyte for *Ectocarpus* studies. Guide RNAs sequence are available in Supplementary Table 8 (see also Fig. 2) and were designed with CRISPOR[76]. Transformation and selection were performed as in refs. 77,78 using Cas12 enzyme (Alt-R L.b. Cas12a /Cpf1, 15.6 µM, IDT) as follows: ribonucleoproteins (RNPs) were assembled by mixing individual crRNAguide (IDT) at a 20 µM volume (EcAPT1-Cas12 guide from ref. 75 and each of the six guides against the flanking regions of EVEc, Supplementary Table 8) with Cas12 enzyme (1 mg ml$^{-1}$ final), in NEB buffer 3.1 (New England Biolabs) (1× final) in IDTE buffer (IDT), incubated for 15 min at room tempereture and conserved at 4 °C until use. Gametophytes of the Ec25 strain were grown at 14 °C in NSW until maturity (4 weeks) and collected on a 50-µm mesh. Gametophytes were maintained in semi-dry conditions in darkness for 3 h at 14 °C, then gamete release was induced by adding ice-cold NSW. Gametes were dilluted to 1$^4$ cells µl$^{-1}$ and 100 µl of gametes were mixed with RNP mix (5 µl of each RNP, 30 µl total) and 130 µl PEG 8000 40% (diluted in NSW) in a Petri dish and left in the dark at room temperature (22 °C) for 30 min before flooding the plates with 20 ml NSW with 0.5PE. Cells were incubated in the dark at 22 °C for 24 h then transferred to 14 °C in low light. Then 2-fluoroadenine was added for selection 48 h after the transformation, at a concentration of 20 µM. Selection was maintained for 30 days with media refreshment every 10 days before isolation of growing germlings. Putative mutants were screened first for the presence of EVEc (genotyping was performed as above with primers for an internal gene of EVEc; Supplementary Table 7), then EVEc-negative algae were further screened with primers spanning the EVEc insertion site. Positive amplicons were sequenced by Sanger sequencing at Azenta (Genewiz) (Supplementary Table 8). An individual showing mutations at the guide RNA sites but still positive for EVEc was kept as control ('APT control'). Viral symptoms were assessed as explained above for the segregating progenies.

### Transmission electron microscopy

Filaments showing morphologically altered plurilocular sporangia were cut and high-pressure frozen (HPF Compact 03, Engineering Office M. Wohlwend GmbH), freeze-substituted (AFS2, Leica Microsystems) with 0.2% OsO$_4$ and 0.1% uranyl acetate in acetone containing 1.5% H$_2$O as substitution medium[79] and embedded in Epon. For transmission electron microscopy, ultrathin sections were stained with uranyl acetate and lead citrate and analysed with a JEM-2100Plus (Jeol) operated at 200 kV.

Viral particle size was measured by averaging the length of three transects for each hexagonal particle section with Fiji[80] (v2.14.0). A total of 42 particles from 4 different sections were measured.

### Viral particle detection by flow cytometry

The presence of viral particles was assessed by flow cytometry. Briefly, viral release was induced as a gamete release[81]: ice-cold ASW was added on semi-dry gametophytes and incubated 1 h at 10 °C. Then 1 ml of media surrounding the gametophytes was collected and spin for 10 min at 11,000*g* at 4 °C. The supernatant was further concentrated on a 300-kDa PES Microcon column (Merck). Next, 10 µl of sample was fixed

with 0.5% gultaraldehyde 15 min at 4 °C, diluted 10 times in TE buffer and stained with SybrGold (Invitrogen) (4× final) for 10 min at 80 °C. Samples were run in a BD FACSMelody cell sorter (BD Bioscience) and the viral population identified using the FITC and SSC channel.

Concentrated viral particle fractions were diluted ten times in water and digested by DNase I (Zymo research) following the manufacturer's instruction to remove non-protected DNA. Then 2.5 μl of digested samples were used for direct PCR (Terra direct PCR kit, Takara), with primers amplifying an EVEc internal gene marker or the algal ITS1 (Supplementary Table 7)

### RNAseq and analysis

Ec25 gametophytes grown at 10 °C or 20 °C for 4 weeks in triplicates were collected on a 40-μm strain and snap-frozen in liquid nitrogen. RNA was extracted using the same protocol as in ref. 77. Library were prepared with the NEBNext Ultra II Directional RNA Library Prep kit for Illumina and sequencing was performed on a NextSeq2000 instrument.

All six samples, three Ec25 gametophytes grown at 10 °C and three at 20 °C, were mapped by HISAT2[82] v2.2.1 with the options '-q --max-intronlen 50000 --passthrough --read-lengths 151' to the EVEc virus genome. Read counts were extracted by featureCounts[83] v2.0.3 additional option '−countReadPairs -s 0 -C −largestOverlap −fraction -O'.

Differentially expressed genes between 10 °C and 20 °C samples were obtained by the R (v4.5.1) package DESeq2[84] (v1.46.0). DEGs were filtered by the adjusted $P$ value threshold of 0.001 and a $|\log_2(\text{fold change})|$ of 1.

### Reporting summary

Further information on research design is available in the Nature Portfolio Reporting Summary linked to this article.

## Data availability

Data are available in Supplementary Information and via the Edmond Repository at https://doi.org/10.17617/3.26JRX4. The raw sequence reads for the Oxford Nanopore data and RNAseq libraries are available via the Sequence Read Archive under BioProject accession number PRJNA1328951. Source data are provided with this paper.

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

## Acknowledgements

This work was supported by the MPG, the ERC (grant no. 864038 to S.M.C.) and the BMBF-funded de.NBI Cloud within the German Network for Bioinformatics Infrastructure (de.NBI) (grant nos. 031A532B, 031A533A, 031A533B, 031A534A, 031A535A, 031A537A, 031A537B, 031A537C, 031A537D and 031A538A). S.M.C. is supported by the Moore Foundation (GBMF11489) and the Bettencourt-Schuller Foundation. R.J.C. was supported by Marie Skłodowska-Curie grant agreement no. 101109906. Computations were also performed in the MPCDF Cobra supercomputer in Garching, Germany, and the cluster of the Max Planck Campus in Tübingen, Germany.

## Author contributions

C.D. performed the investigation, formal analysis, visualization, writing—original draft and writing—review and editing. R.J.C. performed the investigation, visualization, methodology and writing—original draft. C.M. and F.A. performed the investigation. R.L., K.H., P.E. and V.A. performed the investigation and methodology. F.B.H. performed the investigation, formal analysis, visualization and data curation. S.M.C. performed the conceptualization, funding acquisition, methodology, project administration, supervision and writing—review and editing.

## Funding

## Competing interests

The authors declare no competing interests.

## Additional information

**Extended data** is available for this paper at https://doi.org/10.1038/s41564-026-02361-z.

**Correspondence and requests for materials** should be addressed to Susana M. Coelho.

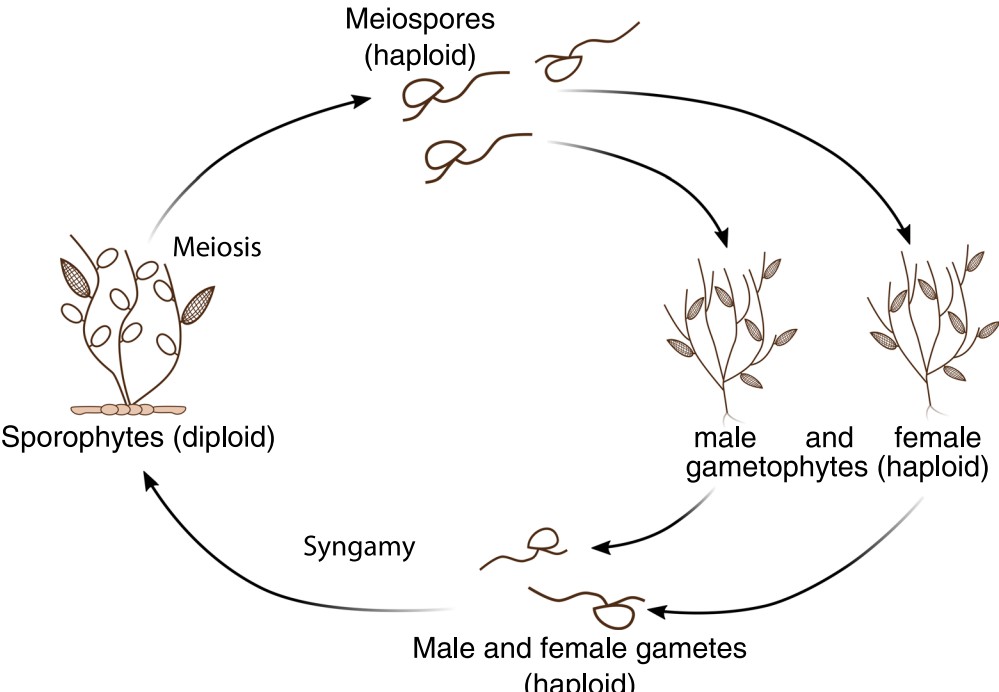

**Extended Data Fig. 1 | Simplified life cycle of *Ectocarpus sp.*** Diploid sporophytes produce unilocular sporangia, where meiosis takes place. *Ectocarpus* possesses a U/V sexual system, where sex is expressed at the haploid stage. Meiospores carrying either the U or the V sex chromosome are released in the environment where they settle to give rise to the female and male haploid gametophytes, respectively. At maturity, gametophytes produce gametes in plurilocular sporangia. Male and female gametes released in the surrounding seawater fuse into diploid zygotes, that in turn develop into the diploid sporophyte.

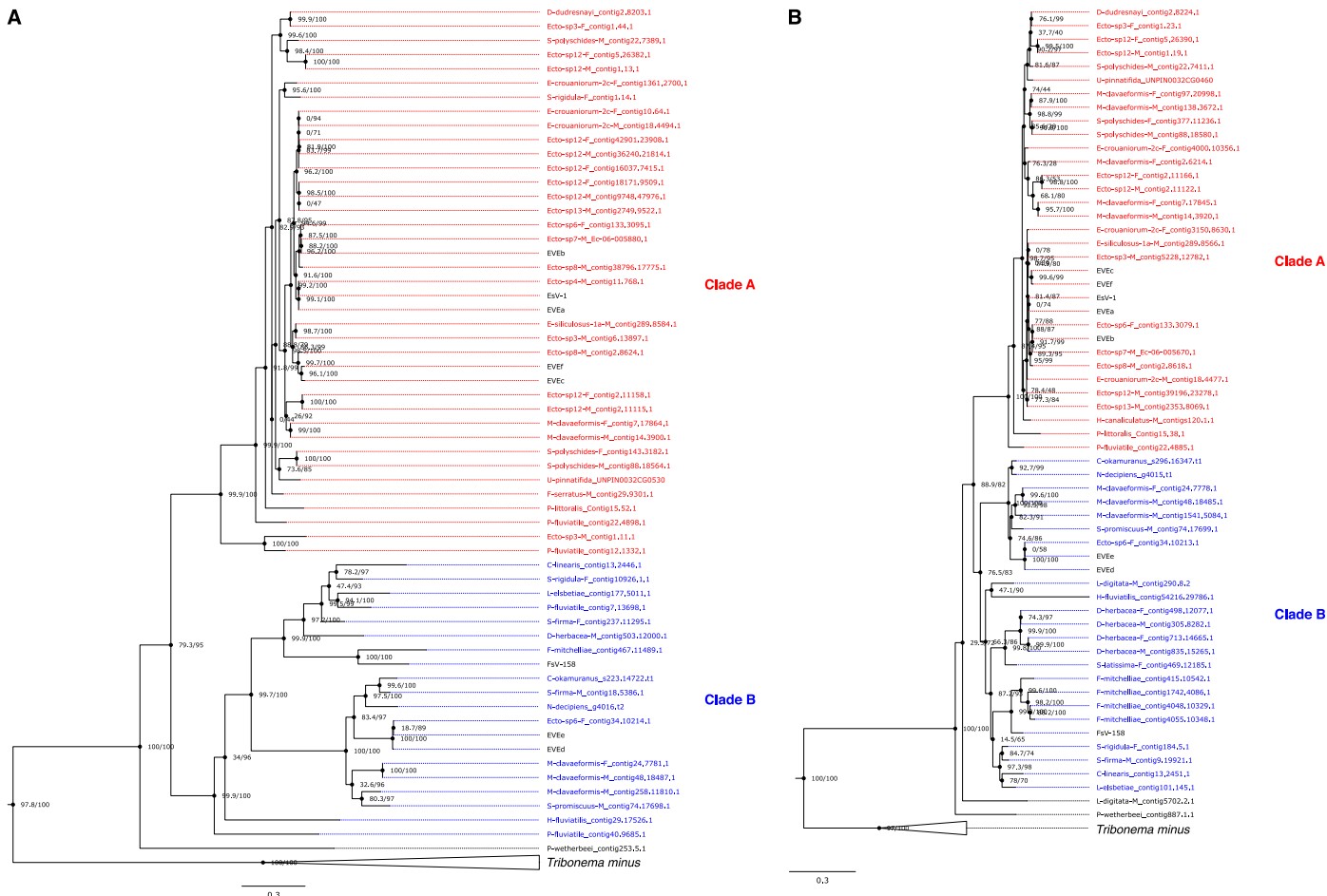

**Extended Data Fig. 2 | DNA Polymerase and Major Capsid Protein phylogenetic trees.** DNA Polymerase (**A**) and Major Capsid Protein (**B**) phylogenetic trees including sequences found in other Phaeophyceae. The trees were rooted with sequence from the Xanthophyte *Tribonema minus* (collapsed clade).

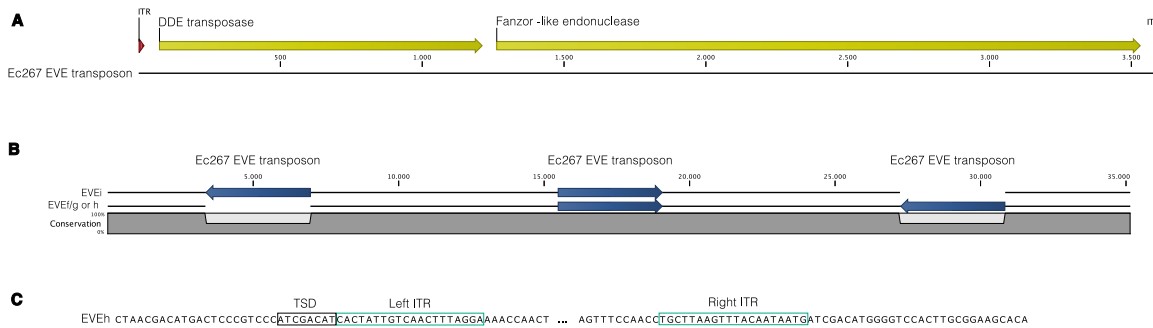

**Extended Data Fig. 3 | Characteristics of the transposable element from EVEf, g, h and i. A**) Structure of the transposon. **B**) Example of two polymorphic/hemizygous sites in different EVEs from Ec267. **C**) Zoom on a polymorphic/hemizygous site showing the 8 bp target site duplication (TSD) and the inverted terminal repeat (ITR) sequences.

```
                    Ec17, chromosome 16
Insertion site    GAGTTCGTAAGGTGCGGTTACAGTTTGTTC---------------------------------------------------------------CCAAGAGCACTCTCTCAACTCTCAACTACA
  (With EVEc)      GAGTTCGTAAGGTGCGGTTACAGTTTGTTCCACCTGTATAACTGTTTTTGCGACACCGCC ... EVEc ... TTAACGAGATTTGTGCAAAACCGGTTATGCCCAAGAGCACTCTCTCAACTCTCAACTACA

                    Ec17, chromosome 26
Insertion site    ACTGGTGAAAAAAAAGAAAGACCAACTAC---------------------------------------------------------------CGGTTCGAGATGGCCACAGCACCACAGCGG
  (With EVEd)      ACTGGTGAAAAAAAAGAAAGACCAACTACCTCATGTATAAAGCAAAACGCGACTTTACC ... EVEd ... TGAACCGGATTGTCGCGTTTTCATTTATCCGGTTCGAGATGGCCACAGCACCACAGCGG

                    Ec17, chromosome 03
Insertion site    GGCCACCCGTGTACTGATCGTTCTTGATTC---------------------------------------------------------------CTTTTACTTTCACTACCGCGCACTTCGAGC
  (With EVEe)      GGCCACCCGTGTACTGATCGTTCTTGATTCCGCATATATTCAGACGTCTGAATGTCAACC ... EVEe ... CCCAGCGGGTGGAGACTGTGGACTCTACACCTTTTACTTTCACTACCGCGCACTTCGAGC

                    Ec17, chromosome 06
Insertion site    TAGTGTGTGTTGAAGATGCTTGTATTTCAA---------------------------------------------------------------CTGTATTTTCGGACTGCAGACGCGACCAAG
  (With EVEb)      TAGTGTGTGTTGAAGATGCTTGTATTTCACCGTCTGAATAACTGTTTTTGCGACACCGCC ... EVEb ... TGAACGGGATTCGGGCGTTTTAATTTATGCCTGTATTTTCGGACTGCAGACGAGATCAAG

                    Ec267, chromosome 23
Insertion site    TGGATCTCCTTGGCACGCGCGGGCAAAAAC---------------------------------------------------------------CGCCCTGCTGCTGCTGCCCTTGTATTCCAA
  (With EVEi)      TGGATCTCCTTGGCACGCGCGGGCAAAAACCTCTTGTATAACTGTTTTTGCGACACCGCC ... EVEi ... TGAGAGGGAATCGTGCAAAACCCCTTATACCGCCCTGCTGCTGCTGCCCTTGTATTCCAA

                    Ec267, chromosome 23, second site
Insertion site    CTCCCCTCCCACTTCTGAGGCCGAACATTC---------------------------------------------------------------CCCCTCTCTCTCACCCTTTTTCGTAGACGC
  (With EVEh)      CTCCCCTCCCACTTCTGAGGCCGAACATTCCTCTTGTATAACTGTTTTTGCGACACCGCC ... EVEh ... TGAGAGGGAATCGTGCAAAACCCCTTATGCCCCCTCTCTCTCACCCTTTTTCGTAGACGC

                    Ec267, chromosome 17
Insertion site    CTCACGGAAAGGCCCTATTTTTGTACATTC---------------------------------------------------------------CATTACAACGCGTGGTTTGACGAATGTTCA
  (With EVEg)      CTCACGGAAAGGCCCTATTTTTGTACATTCCTCTTGTATAACTGTTTTTGCGACACCGCC ... EVEg ... TGAGAGGGAATCGTGCAAAACCCCTTATGCCATTACAACGCGTGGTTTGACGAATGTTCA

                    Ec267, chromosome 15
Insertion site    TTCGTAGGGTGCTGGGTCTTCTCTTGCTAC---------------------------------------------------------------CGCATTCCGTGTGGCGTTCGTCCTGCACCC
  (With EVEf)      TTCGTAGGGTGCTGGGTCTTCTCTTGCTACCTGTTGTATAACTGTTTTTGCGACACCGCC ... EVEf ... TGAGAGGGAATCGTGCAAAACCCCTTATGCCGCATTCCGTGTGGCGTTCGTCCTGCACCC
```

**Extended Data Fig. 4 | Characteristics fo the insertion site of EVEb to EVEi.** Insertion sites sequences of EVEb - i., highlighting in red the conserved CC dinucleotide in EVE-free chromosome insertion sites, and flanking each EVE insertion.

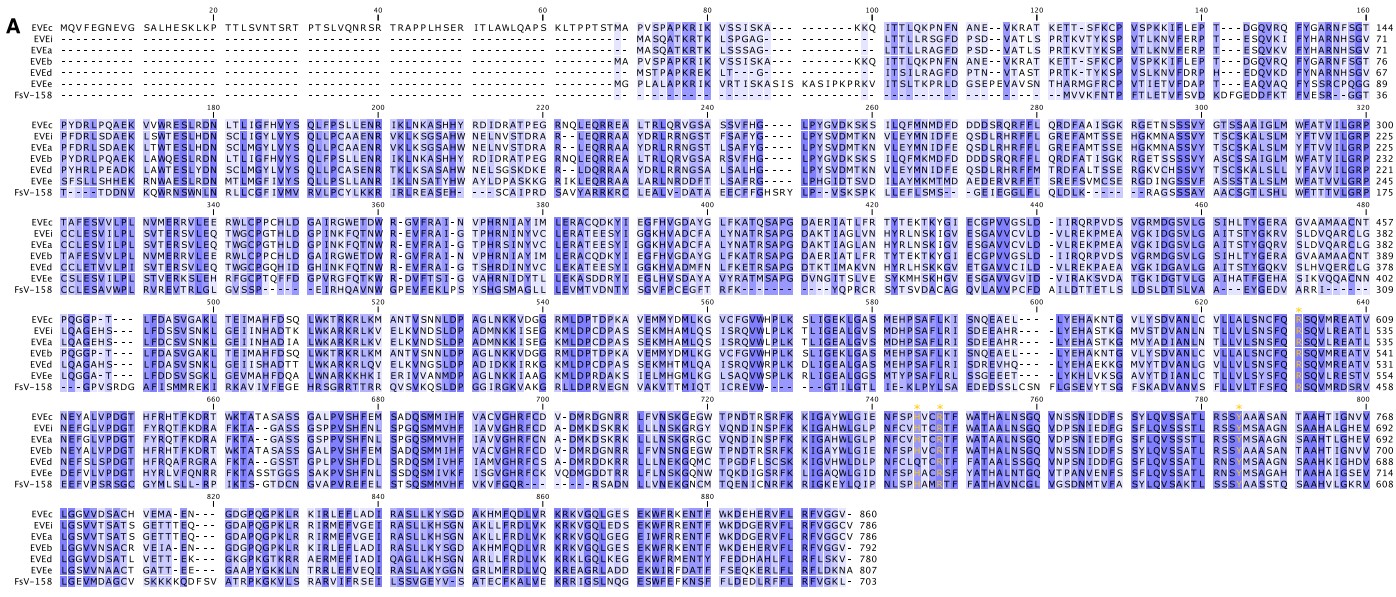

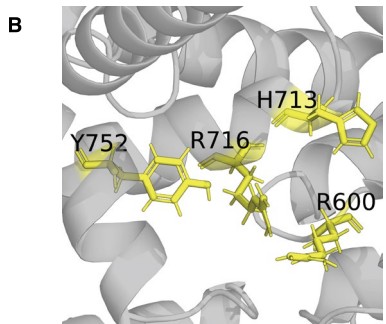

**Extended Data Fig. 5 | Characteristics of the tyrosine recombinase. A)** Alignment of the EVE and phaeoviral tyrosine recombinases; putative catalytic residues are highlighted in yellow. **B)** Catalytic site of EVEc tyrosine recombinase with the putative catalytic residues highlighted in yellow.

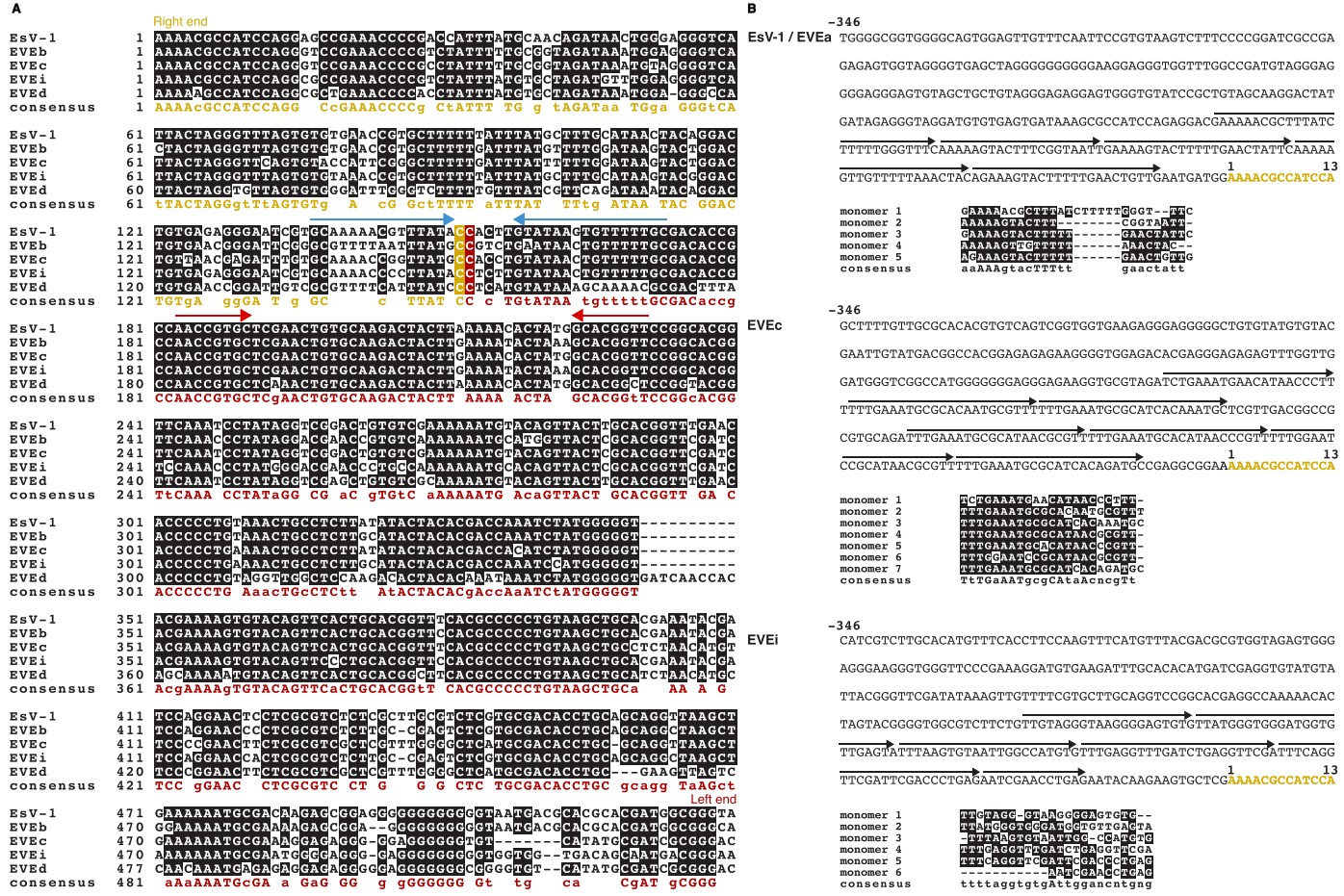

**Extended Data Fig. 6 | Characteristics of the *att*P site. (A)**, alignment of the conserved part of the attP site, with the right end termini (in yellow on the consensus sequence) and left end termini (in red) flanking the central CC insertion site. Red arrows highlight conserved inverted repeat in the left end, and blue arrow the inverted repeats flanking the CC insertion site. **(B)**, Direct repeats upstream of the right end sequence presented in A, with arrows presenting the position of the repeats, and their alignment. Yellow letters indicate the start of the sequence in A.

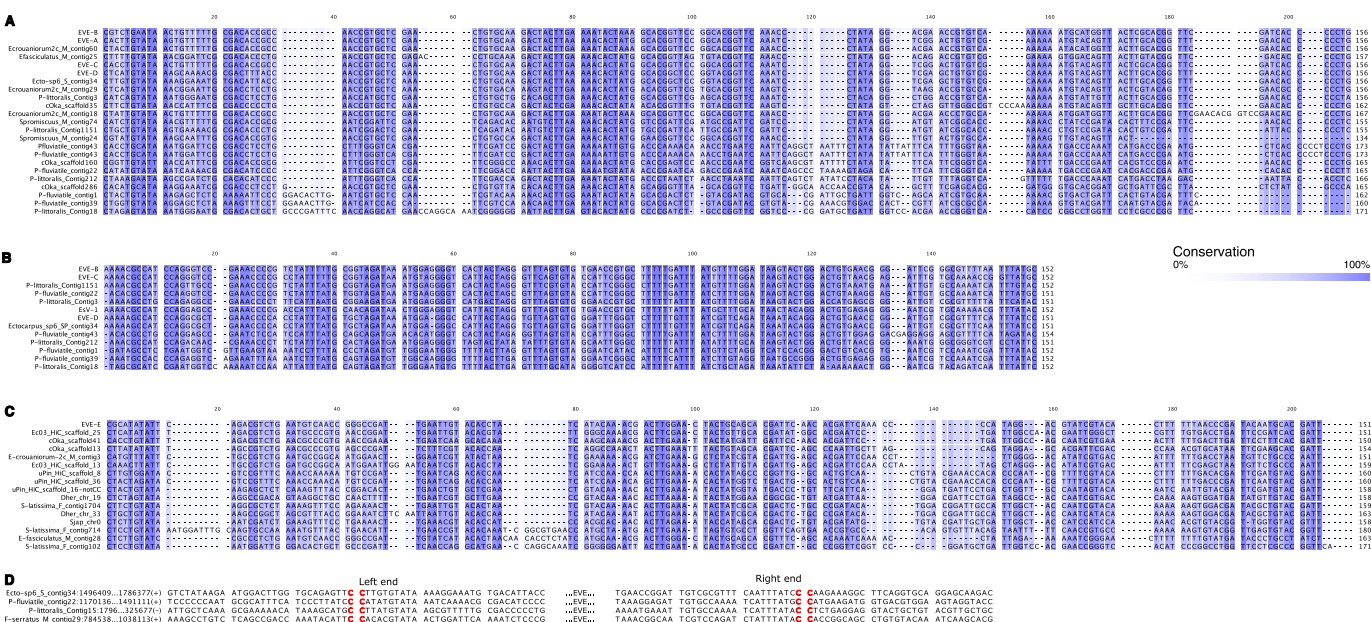

**Extended Data Fig. 7 | Alignment of the termini of the EVEs identified here and putative EVE termini from other representative Phaeophyceae genomes.** (**A**) Left end of EVEc-type termini, (**B**) Right end of EVEc-type termini, (**C**) Left end of EVEe-type termini. (**D**) Boundaries of the full-length EVEs identified in other brown algal genomes[10].

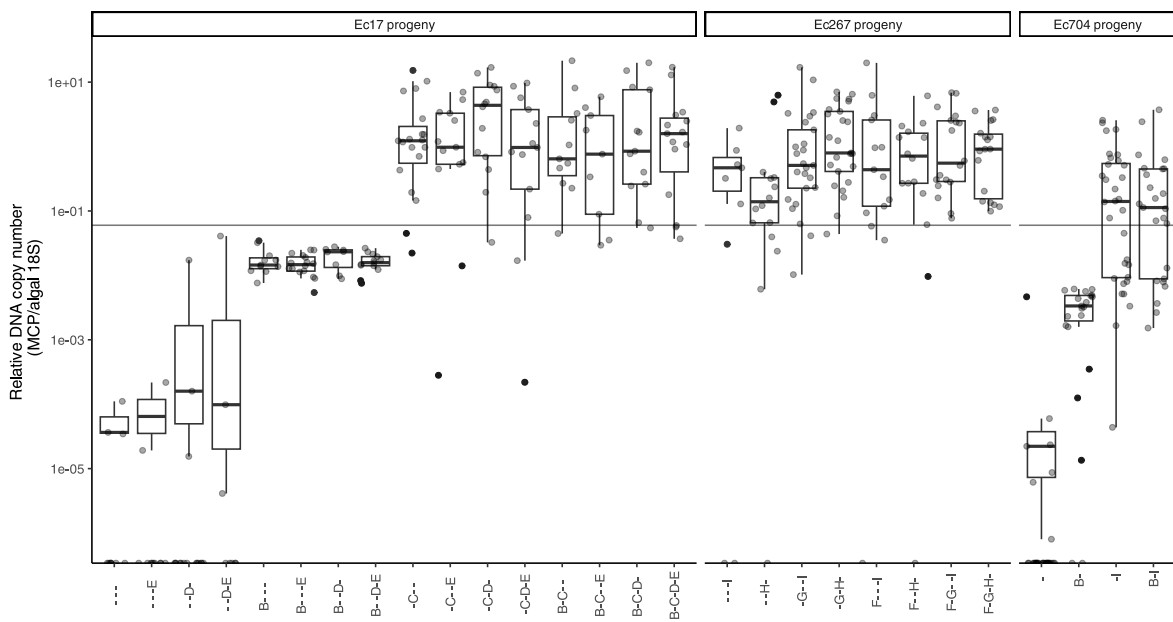

**Extended Data Fig. 8 | qPCR quantification of symptoms in the different progenies generated here as a function of the EVE content.** The horizontal dashed line is the threshold for symptomatic diagnostic, as established by comparing symptoms by microscopy and qPCR in the Ec17 progeny. Each dot corresponds to one individual gametophyte, and the boxplots represents the median of all individuals carrying the same EVE content, with the minima and maxima of the box corresponding to the first and third quartiles, and whiskers to 1.5 time the inter-quartile range.

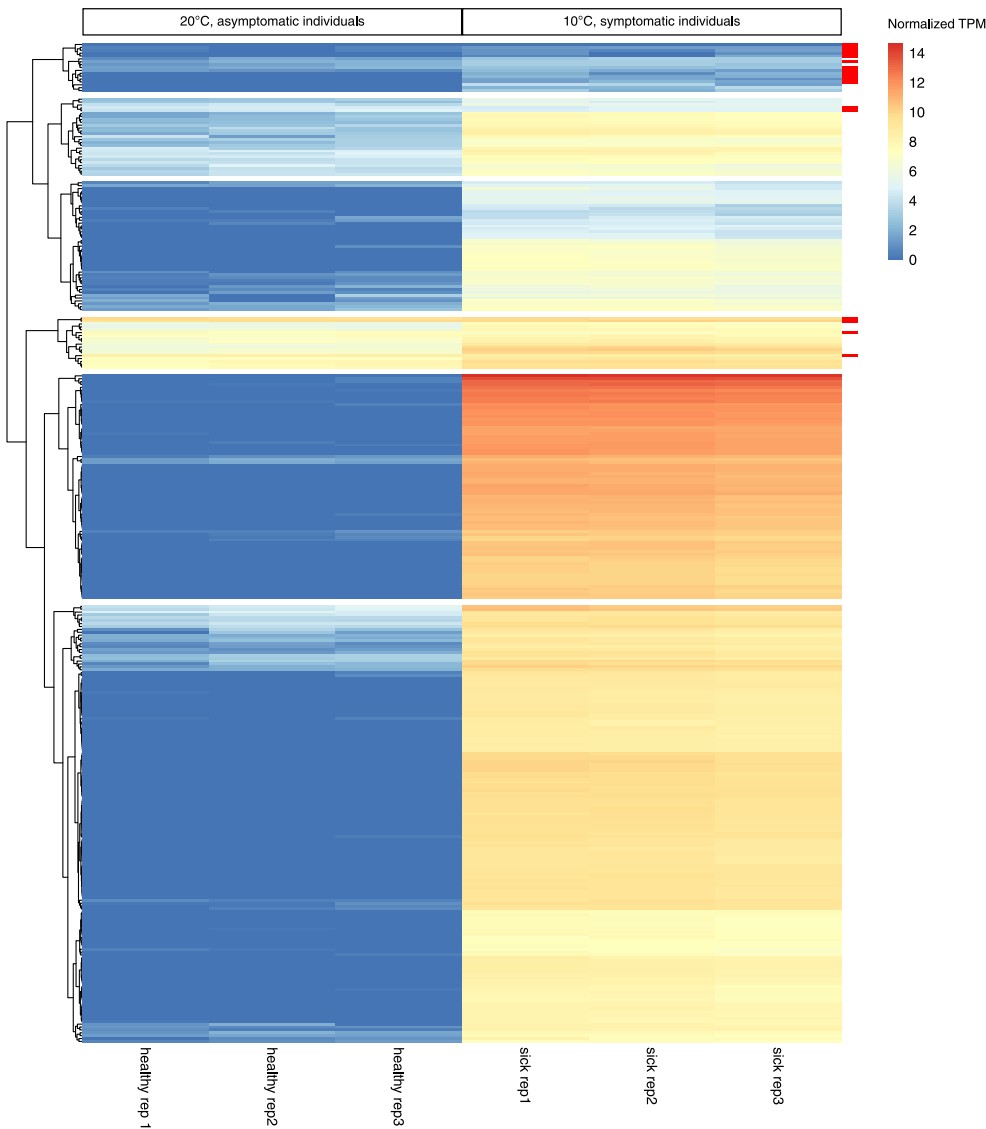

**Extended Data Fig. 9 | RNA-Seq analysis of EVEc expression in symptom-inducing (sick, 10 °C) and non-inducing (healthy, 20 °C) conditions.** heatmap of min-max normalized transcripts count for each EVEc gene in each of the 3 biological replicates. All genes are significantly induced in virus-inducing conditions, that is TPM Fold Change>2 and Wald two-sided test adjusted p-value < 0.05 (default DESeq2 test, [84]) except the ones specifically marked in red.

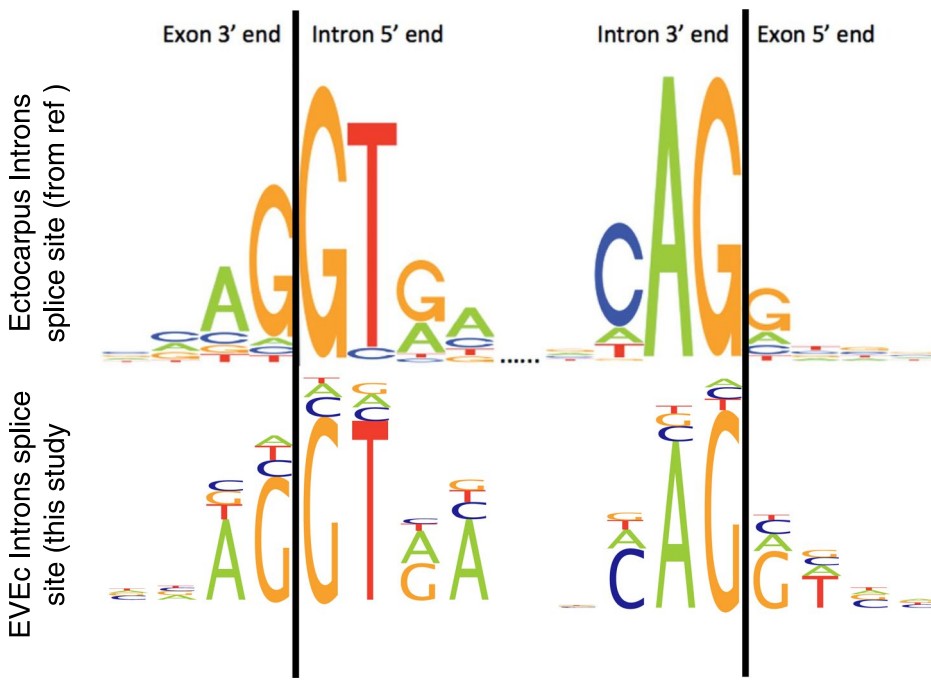

**Extended Data Fig. 10 | EVEc genes carry sliceosomal introns.** Comparison of the sequences of the splicing sites of EVEc genes (bottom) compared with *Ectocarpus* splicing sites (top, from[35]).

# Reporting Summary

## Statistics

For all statistical analyses, confirm that the following items are present in the figure legend, table legend, main text, or Methods section.

| n/a | Confirmed | |
|---|---|---|
| ☐ | ☒ | The exact sample size (*n*) for each experimental group/condition, given as a discrete number and unit of measurement |
| ☐ | ☒ | A statement on whether measurements were taken from distinct samples or whether the same sample was measured repeatedly |
| ☐ | ☒ | The statistical test(s) used AND whether they are one- or two-sided<br>*Only common tests should be described solely by name; describe more complex techniques in the Methods section.* |
| ☐ | ☒ | A description of all covariates tested |
| ☐ | ☒ | A description of any assumptions or corrections, such as tests of normality and adjustment for multiple comparisons |
| ☐ | ☒ | A full description of the statistical parameters including central tendency (e.g. means) or other basic estimates (e.g. regression coefficient) AND variation (e.g. standard deviation) or associated estimates of uncertainty (e.g. confidence intervals) |
| ☐ | ☒ | For null hypothesis testing, the test statistic (e.g. *F*, *t*, *r*) with confidence intervals, effect sizes, degrees of freedom and *P* value noted<br>*Give P values as exact values whenever suitable.* |
| ☒ | ☐ | For Bayesian analysis, information on the choice of priors and Markov chain Monte Carlo settings |
| ☒ | ☐ | For hierarchical and complex designs, identification of the appropriate level for tests and full reporting of outcomes |
| ☒ | ☐ | Estimates of effect sizes (e.g. Cohen's *d*, Pearson's *r*), indicating how they were calculated |

*Our web collection on statistics for biologists contains articles on many of the points above.*

## Software and code

Policy information about availability of computer code

| | |
|---|---|
| Data collection | Minknown 6.8.11 for Oxford Nanopore sequencing, CFX Real-time PCR system (Biorad) for qPCR, NextSeq2000 for Illumina library sequencing (RNASeq), BD FACSMelody cell sorter for flow cytometry, JEM-2100Plus (Jeol) for TEM |
| Data analysis | The programs used for analysis : canu v2.2; Oxford Nanopore's Dorado, v0.6.2; Viralrecall v2.1; minimap2 v2.28;  Medaka v2.0.1; Prodigal (v.2.6.3); Prodigal-gv (v.2.11.0-gv); GeneMarkS-2 (v.1.14_1.25_lic); ORForise (v.1.4.2); JBrowse2; US-align ; Orthofinder (version 2.5.5); HMMER v3.1b2 and 3.4; MAFFT v7.525 ; trimAL v1.2rev59; IQtree (v1.6.12); PyMOL v.2.5.0; AlphaFold (v.2.3.1); WebLogo3<br>Fiji (v2.14.0) for image analysis, FlowJo 10.5.3 for  visualization of the flow cytometry data<br>R 4.3.0  and 4.5.1 with ggplot2 3.5.1 and DESeq2 v1.46.0 packages were used for statistical computing and graphics. |

For manuscripts utilizing custom algorithms or software that are central to the research but not yet described in published literature, software must be made available to editors and reviewers. We strongly encourage code deposition in a community repository (e.g. GitHub). See the Nature Portfolio guidelines for submitting code & software for further information.

# Data

Policy information about [availability of data](availability of data)
  All manuscripts must include a [data availability statement](data availability statement). This statement should provide the following information, where applicable:
    - Accession codes, unique identifiers, or web links for publicly available datasets
    - A description of any restrictions on data availability
    - For clinical datasets or third party data, please ensure that the statement adheres to our [policy](policy)

Data is available as supplementary datasets and in the Edmond Repository https://doi.org/10.17617/3.26JRX4 . The raw sequence reads for the Oxford Nanopore data and RNA-seq libraries are available in the Sequence Read Archive under BioProject accession number PRJNA1328951.

# Research involving human participants, their data, or biological material

Policy information about studies with [human participants or human data](human participants or human data). See also policy information about [sex, gender (identity/presentation), and sexual orientation](sex, gender (identity/presentation), and sexual orientation) and [race, ethnicity and racism](race, ethnicity and racism).

| | |
|---|---|
| Reporting on sex and gender | Use the terms sex (biological attribute) and gender (shaped by social and cultural circumstances) carefully in order to avoid confusing both terms. Indicate if findings apply to only one sex or gender; describe whether sex and gender were considered in study design; whether sex and/or gender was determined based on self-reporting or assigned and methods used. Provide in the source data disaggregated sex and gender data, where this information has been collected, and if consent has been obtained for sharing of individual-level data; provide overall numbers in this Reporting Summary.  Please state if this information has not been collected. Report sex- and gender-based analyses where performed, justify reasons for lack of sex- and gender-based analysis. |
| Reporting on race, ethnicity, or other socially relevant groupings | Please specify the socially constructed or socially relevant categorization variable(s) used in your manuscript and explain why they were used. Please note that such variables should not be used as proxies for other socially constructed/relevant variables (for example, race or ethnicity should not be used as a proxy for socioeconomic status). Provide clear definitions of the relevant terms used, how they were provided (by the participants/respondents, the researchers, or third parties), and the method(s) used to classify people into the different categories (e.g. self-report, census or administrative data, social media data, etc.) Please provide details about how you controlled for confounding variables in your analyses. |
| Population characteristics | Describe the covariate-relevant population characteristics of the human research participants (e.g. age, genotypic information, past and current diagnosis and treatment categories). If you filled out the behavioural & social sciences study design questions and have nothing to add here, write "See above." |
| Recruitment | Describe how participants were recruited. Outline any potential self-selection bias or other biases that may be present and how these are likely to impact results. |
| Ethics oversight | Identify the organization(s) that approved the study protocol. |

Note that full information on the approval of the study protocol must also be provided in the manuscript.

# Field-specific reporting

Please select the one below that is the best fit for your research. If you are not sure, read the appropriate sections before making your selection.

☐ Life sciences      ☐ Behavioural & social sciences      ☒ Ecological, evolutionary & environmental sciences

For a reference copy of the document with all sections, see [nature.com/documents/nr-reporting-summary-flat.pdf](nature.com/documents/nr-reporting-summary-flat.pdf)

# Ecological, evolutionary & environmental sciences study design

All studies must disclose on these points even when the disclosure is negative.

| | |
|---|---|
| Study description | Vertical transmission of viral infection in Ectocarpus sp, linked to the viral genomes insertions in the algal genomes |
| Research sample | Ectocarpus species 7  sporophytes showing viral symptoms or viral insertion in their genomes and available in culture collections. More details on the origin of the different algae are available in supplementary table 1 |
| Sampling strategy | Ectocarpus species 7  sporophytes suspected to have a link with viral infection and available in culture collections were used to generate > 100 gametophyte porgeny for each sporophyte |
| Data collection | Each gametophyte was studied for viral symptoms and for the presence of viral insertions in its genome by microscopy and qPCR (for symptoms) and PCR (genotyping) |
| Timing and spatial scale | not applicable |

| Data exclusions | no data were excluded |
|---|---|
| Reproducibility | More than 100 individual gametophytes were generated for each sporophyte, allowing a good statistical power. Each inidividual was examined once by microscopy and PCR while the qPCR was performed in technical triplicates |
| Randomization | Phenotyping for viral symptoms was done in a random order with no a priori on the health status of the algae. |
| Blinding | Phenotyping for viral symptoms was done blindly with no a priori on the health status of the algae. Identically, PCR genotyping and qPCR was done without knowing the health stauts of the algae. |

Did the study involve field work? ☐ Yes ☒ No

# Reporting for specific materials, systems and methods

We require information from authors about some types of materials, experimental systems and methods used in many studies. Here, indicate whether each material, system or method listed is relevant to your study. If you are not sure if a list item applies to your research, read the appropriate section before selecting a response.

## Materials & experimental systems

| n/a | Involved in the study |
|---|---|
| ☒ | ☐ Antibodies |
| ☒ | ☐ Eukaryotic cell lines |
| ☒ | ☐ Palaeontology and archaeology |
| ☒ | ☐ Animals and other organisms |
| ☒ | ☐ Clinical data |
| ☒ | ☐ Dual use research of concern |
| ☒ | ☐ Plants |

## Methods

| n/a | Involved in the study |
|---|---|
| ☒ | ☐ ChIP-seq |
| ☐ | ☒ Flow cytometry |
| ☒ | ☐ MRI-based neuroimaging |

## Plants

| Seed stocks | *Report on the source of all seed stocks or other plant material used. If applicable, state the seed stock centre and catalogue number. If plant specimens were collected from the field, describe the collection location, date and sampling procedures.* |
|---|---|
| Novel plant genotypes | *Describe the methods by which all novel plant genotypes were produced. This includes those generated by transgenic approaches, gene editing, chemical/radiation-based mutagenesis and hybridization. For transgenic lines, describe the transformation method, the number of independent lines analyzed and the generation upon which experiments were performed. For gene-edited lines, describe the editor used, the endogenous sequence targeted for editing, the targeting guide RNA sequence (if applicable) and how the editor was applied.* |
| Authentication | *Describe any authentication procedures for each seed stock used or novel genotype generated. Describe any experiments used to assess the effect of a mutation and, where applicable, how potential secondary effects (e.g. second site T-DNA insertions, mosiacism, off-target gene editing) were examined.* |

## Flow Cytometry

### Plots

Confirm that:

☒ The axis labels state the marker and fluorochrome used (e.g. CD4-FITC).

☒ The axis scales are clearly visible. Include numbers along axes only for bottom left plot of group (a 'group' is an analysis of identical markers).

☒ All plots are contour plots with outliers or pseudocolor plots.

☒ A numerical value for number of cells or percentage (with statistics) is provided.

### Methodology

| Sample preparation | 0.5% gultaraldehyde fixation followed by Sybrgold staining (final concentration 4X) |
|---|---|
| Instrument | BD FACSMelody cell sorter (BD Bioscience) |
| Software | FlowJo 10.5.3 |
| Cell population abundance | 4 000 000 events were measured for each sample |

Gating strategy   The gating strategy was used only to remove the background noise. This is more a qualitative than quantitative analysis.

☒ Tick this box to confirm that a figure exemplifying the gating strategy is provided in the Supplementary Information.

