## [Peer Review File · Nature Microbiology]

Latent endogenous giant viruses drive active infection and inheritance in a multicellular algal host

Corresponding Author: Dr Susana Coelho

Version 0:

Decision Letter:

16th January 2026

Dear Dr Coelho,

Thank you for your patience while your manuscript "Latent endogenous giant viruses drive active infection and inheritance in a multicellular host" was under peer-review at Nature Microbiology. It has now been seen by 3 referees, whose expertise and comments you will find at the end of this email. Although they find your work of some potential interest, they have raised a number of concerns that will need to be addressed before we can consider publication of the work in Nature Microbiology.

In particular, we would like Reviewer 2's point regarding whether the sense–antisense long-read mapping at the telRL site reflects linear genomes or represents transient replication intermediates addressed either experimentally or detailed in your response to referees' document. In addition to this, Reviewer 3 raised a question regarding hemizygous integrations of EVEs which we would also like to see addressed in the revised manuscript. Aside from these, the remaining comments are quite straightforward.

Should further experimental data allow you to address these criticisms, we would be happy to look at a revised manuscript.

Please include a data availability statement as a separate section after Methods but before references, under the heading "Data Availability". This section should inform readers about the availability of the data used to support the conclusions of your study. This information includes accession codes to public repositories (data banks for protein, DNA or RNA sequences, microarray, proteomics data etc...), references to source data published alongside the paper, unique identifiers such as URLs to data repository entries, or data set DOIs, and any other statement about data availability. At a minimum, you should include the following statement: "The data that support the findings of this study are available from the corresponding author upon request", mentioning any restrictions on availability. If DOIs are provided, we also strongly encourage including these in the Reference list (authors, title, publisher (repository name), identifier, year). For more guidance on how to write this section please see: <http://www.nature.com/authors/policies/data/data-availability-statements-data-citations.pdf>

* If you have not done so already we suggest that you begin to revise your manuscript so that it conforms to our Article format instructions at <http://www.nature.com/nmicrobiol/info/final-submission>. Refer also to any guidelines provided in this letter.

* Include a revised version of any required reporting checklist. It will be available to referees (and, potentially, statisticians) to aid

in their evaluation if the manuscript goes back for peer review. A revised checklist is essential for re-review of the paper.

EXTENDED DATA FIGURES

Link Redacted

Note: This url links to your confidential homepage and associated information about manuscripts you may have submitted or be reviewing for us. If you wish to forward this e-mail to co-authors, please delete this link to your homepage first.

Nature Microbiology is committed to improving transparency in authorship. As part of our efforts in this direction, we are now requesting that all authors identified as 'corresponding author' on published papers create and link their Open Researcher and Contributor Identifier (ORCID) with their account on the Manuscript Tracking System (MTS), prior to acceptance. This applies to primary research papers only. ORCID helps the scientific community achieve unambiguous attribution of all scholarly contributions. You can create and link your ORCID from the home page of the MTS by clicking on 'Modify my Springer Nature account'. For more information please visit www.springernature.com/orcid.

If you wish to submit a suitably revised manuscript we would hope to receive it within 3 months. If you cannot send it within this time, please let us know. We will be happy to consider your revision, even if a similar study has been accepted for publication at Nature Microbiology or published elsewhere (up to a maximum of 3 months).

Yours sincerely,

[redacted]
Associate Editor
Nature Microbiology

Reviewer Expertise:

- Referee #1: endogenous giant viruses, EVEs
- Referee #2: giant DNA viruses, virus-host dynamics
- Referee #3: evolutionary biology and ecology

Reviewer Comments:

Reviewer #1 (Remarks to the Author):

This study shows that the endogenous viral elements in brown algae are latent viruses that can lead to virus production under specific environmental conditions and developmental stages. Authors elegantly demonstrate this using long read sequencing, microscopy and genetics to provide mechanistic basis of free virus production in brown algae. In other words, they establish a 'cause-effect' relationship between the integrated viruses and the produced free virus particles. I believe this is an important research towards understanding the broader consequences of giant virus endogenization in diverse eukaryotes.

I think the research is solid, and have no major concerns regarding the methods or interpretation of the results. I only have a few minor comments:

- a) The authors stated "Giant virus transposons are typically of prokaryotic origin" - I think this is a swipping statement. I agree that certain MGEs in giant viruses likely originated from prokaryotes, but the authors should be careful not to generalize that

all/most GV transposons are of prokaryotic origin.

b) Line 529: Please provide a brief description of this method - just citing the article is not sufficient in my opinion.

c) Same for line 558 in methods. Please provide more details.

d) Line 422-423 : Please add the parameters, specifically the score cutoff and other parameters.

e) Line 471: "previous data" - please elaborate and provide citations if necessary.

f) Line 509: 'hmm model ' should be 'HMM' model. this should be corrected throughout the manuscript where needed.

g) There is no Data availability statement or link to the relevant data, for example the RNA-seq dataset.

Reviewer #2 (Remarks to the Author):

Nucleocytoviruses have been shown to integrate their genomes into those of their eukaryotic hosts, giving rise to distinct genomic fragments termed Giant Endogenous Viral Elements (GEVEs). These elements have been shown to be prevalent among unicellular eukaryotes, and other recent work in green algae has shown that GEVEs represent a latent stage in the viral infection cycle. This study presents another example of a GEVE capable of establishing an active infection, however, for the first time in a model multicellular organism. Using long-read sequencing, the authors resolved multiple phaeovirus viral insertions, ranging from 288 to 407 Kbp in size, across multiples *Ectocarpus* lines. They show that specific host developmental stages and environmental cues lead to the onset of viral symptoms and subsequent viral particle production. Genotyping of the haploid symptomatic and asymptomatic progeny from different lines revealed specific viral insertions that were putatively active and later shown to be transcriptionally reactivated under cold treatment. Active elements do not seem to interact with or change the severity of their associated symptoms when present alongside other inactive elements. Finally, analysis of the genomic content of these GEVEs, together with characterization of their end termini, supports a model in which viral genome integration and excision occur via a tyrosine recombinase interacting with attB/attP -like sites, paired with genome linearization mediated by a telomerase resolvase.

Overall, this is a well-written and meticulously performed study that highlights an exciting system to explore latency in nucleocytoviruses of multicellular organisms. Although, phaeovirus signatures have been shown to be prevalent among brown algal genomes, the role of latency in their infection cycles has remained unclear. This study provides strong evidence for new integration and reactivation dynamics of GEVEs and highlights the importance of host biology and environmental context in modulating the reactivation of integrated phaeoviruses in natural settings.

One unresolved question in the manuscript that was highlighted in the discussion, is whether the sense–antisense long-read mapping at the telRL site reflects linear genomes with covalently closed telomeres or instead represents transient replication intermediates of the viral genome. Long-read sequencing of the purified viral particles DNA could help resolve if the linear form is packaged in mature virions. An assessment of the proportion of reads spanning the attP site and those showing sense–antisense mapping at the telRL site, ideally in a time-series during induction, could help resolve the sequence of events during viral reactivation. We understand if these experiments are outside of the scope of this manuscript.

Minor comments

In the abstract I would shy away from statements of priority (“the first direct evidence” could just be “direct evidence”) – some journals advise against these statements given the nuance often involved in these statements, and I don’t think it is needed here given the clear advances presented. I do not think that mitigates the importance of this work at all.

The text on introns is intriguing. Spliceosomal introns have also been found in chloroviruses, and it would be worth citing that here for context – Fitzgerald et al *Jvi* doi: 10.1016/j.virol.2006.08.033 and Sun et al *Journal of Molecular Evolution* <https://doi.org/10.1007/s002399910009>

P.2 Ln.51: The FirrV-1 genome has been partially sequenced. I would suggest indicating that only these two genomes have been sequenced in their entirety.

P.4 Ln.129: In consistency with Fig. S6 and Fig. S7 termini names, should the “CC” dinucleotide form part of the right terminus instead?

P.6 Ln.197: Text indicate that 110 haploid individuals were obtained for line Ec267, but Fig. 2A shows n=140.

P.7 Ln. 218: To help in the interpretation of Fig. 2G, I would suggest clarifying briefly what constitute as viral symptoms according to the microscopy results. Indicate in the caption of Fig. 2G that these were observed in the gametangia.

P.8. Ln.264,266: Figure 3C refers to the TEM and not the telRL site.

P.9 Ln. 279: Should it be Fig. 2B,G?

P.28 Fig.S2: In A) EVEb is missing, and in B) EVEc is misspelled.

P.29 Fig.S3: Keep consistency in the caption and figure, either ITR or TIR.

Typos and changes for consistency:

P.1 Ln.17: do not capitalize phaeoviruses if referred to as a collective group of viruses. Check for other instance throughout the text.

P.2 Ln.41: Italicize Nucleocytoviricota. Check for other instance throughout the text.

Ln.163: do not capitalize xanthoviruses if referred to as a collective group of viruses. Check for other instance throughout the text.

P.10 Ln.332: do not capitalize punuiviruses if referred to as a collective group of viruses. Check for other instance throughout the text.

text.

- P.13 Ln. 414: add space in "100µm".
- P.13 Ln. 417: change "grinded" to "ground".
- P.13 Ln. 420: change "RNAseA" to "RNase A".
- P.13 Ln. 421: "µLProteinaseK" to "µL Proteinase K". Indicate Proteinase K concentration.
- P.13 Ln. 422: add space in "200µL". Change "RNAseA" to "RNase A".
- P.13 Ln. 423: add space in "1.5h".
- P.13 Ln. 427: change "Qbit" to "Qubit".
- P.14 Ln. 460: change "RNA-seq" to "RNA-Seq".
- P.14 Ln. 469: change "trimal" to "trimAl".
- P.14 Ln. 470: change "IQtree" to "IQ-TREE".
- P.14 Ln.472: add space in "DNAPolymerase".
- P.14 Ln.478: change "Tree" to "The tree".
- P.15 Ln.519: change "cd-hit" to "CD-HIT".
- P.15 Ln. 521: change "clustalo, trimmed with trimAL" to "ClustalO, and trimmed with trimAl".
- P.16 Ln. 534: change "Material" to "The material".
- P.16 Ln.537-538: change "grinded" to "ground". Change "for two times" to "twice for".
- P.16 Ln.541: change "grinded" to "ground".
- P.16 Ln.545: change "40 cycles" to "40 cycles of amplification".
- P.16 Ln.550: change "PCR were performed on the same grinded algal" to "PCR was performed on the same ground algal".
- P.16 Ln.552: I suggest enclosing "see Table S7 for the amplification cycle programs" in parenthesis.
- P.16 Ln.557: change "See" to "see".
- P.17 Ln.585: change "DNAse" to "DNase".
- P.17 Ln.588: missing a period.
- P.27 Ln.795-796: misspelled "meiosis" and "Meiospores".

Reviewer #3 (Remarks to the Author):

Duchene et al. present a comprehensive study demonstrating the presence of an active giant provirus in a multicellular host. This is quite groundbreaking and exciting, as this type of viral lifestyle has so far been attributed only to smaller viruses and mostly in unicellular hosts. The findings, thus, open a new window into viral and microbial ecology with far-reaching implications.

I found the paper to be well-written and clear. The different aspects of the findings are analyzed comprehensively, using complementary approaches and angles, and in my opinion, provide solid support for the reported findings. This includes thorough genomic analysis, a convincing hypothesis for the integration mechanism, and a compelling demonstration of temperature-induced activation.

I only have a few minor comments/questions

Diploid assemblies with hemizygous integrations can be tricky. In particular, exiting hemizygous integration in one locus can be missed if the same integration also already exists at a different locus. Such an effect could be checked by investigating the depth of read coverage between EVEs and host contigs. In case of a single, hemizygous integration, EVE coverage should be half that of the mean host coverage. Higher coverage of the EVE could indicate the presence of additional, unobserved integrations

According to the methods, the authors grew the hosts for a relatively long time under different temperature conditions (4 weeks at 10°C) before analyzing viral activity via RNA seq. I was wondering if it also takes that long to observe symptoms after giving the trigger, or if it is also possible to trigger viral production in much shorter periods of coldness, and if it would be even possible to detect exactly when activation starts?

There might be a typo in the M&M: MCsan > MCscan?

Version 1:

Decision Letter:

Our ref: NMICROBIOL-25103697A

12th March 2026

Dear Dr. Coelho,

Thank you for submitting your revised manuscript "Latent endogenous giant viruses drive active infection and inheritance in a multicellular host" (NMICROBIOL-25103697A). It has now been seen by the original referees and their comments are below. The reviewers find that the paper has improved in revision, and therefore we'll be happy in principle to publish it in Nature Microbiology, pending minor revisions to satisfy the referees' final requests and to comply with our editorial and formatting guidelines.

Thank you again for your interest in Nature Microbiology Please do not hesitate to contact me if you have any questions.

Sincerely,

[redacted]
Associate Editor
Nature Microbiology

Reviewer #1 (Remarks to the Author):

Thank you for addressing my suggestions. I have no further comments.

Reviewer #2 (Remarks to the Author):

The authors have done an excellent job responding to comments, and this will be an outstanding contribution to the literature. Congrats to the authors!

Reviewer #3 (Remarks to the Author):

In my opinion, the authors have adequately addressed all concerns raised by the reviewer. I support publication of the article in its current form.

Version 2:

Decision Letter:

15th April 2026

Dear Dr Coelho,

I am pleased to accept your Article "Latent endogenous giant viruses drive active infection and inheritance in a multicellular algal host" for publication in Nature Microbiology. Thank you for having chosen to submit your work to us and many congratulations.

After the grant of rights is completed, you will receive a link to your electronic proof via email with a request to make any corrections within 48 hours. If, when you receive your proof, you cannot meet this deadline, please inform us at rjsproduction@springernature.com immediately. You will not receive your proofs until the publishing agreement has been

received through our system

Authors may need to take specific actions to achieve compliance with funder and institutional open access mandates. If your research is supported by a funder that requires immediate open access (e.g. according to [Plan S principles](https://www.springernature.com/gp/open-science/plan-s-compliance) or the [NIH public access policy](https://www.springernature.com/gp/open-science/us-federal-agency-compliance)) then you should select the gold OA route, and we will direct you to the compliant route where possible. Because authors warrant under our subscription licensing terms that they haven't committed to licensing any version of their article under a licence inconsistent with the terms of our agreement – including the applicable embargo period – publication under the subscription model isn't suitable for authors whose funders require no embargo.

With kind regards,

[redacted]
Associate Editor
Nature Microbiology

P.S. Click on the following link if you would like to recommend Nature Microbiology to your librarian
<http://www.nature.com/subscriptions/recommend.html#forms>

** Visit the Springer Nature Editorial and Publishing website at http://editorial-jobs.springernature.com?utm_source=ejp_NMicro_email&utm_medium=ejp_NMicro_email&utm_campaign=ejp_NMicro for more information about our career opportunities. If you have any questions please click [here](mailto:editorial.publishing.jobs@springernature.com).

Dear Dr Coelho,

Thank you for your patience while your manuscript "Latent endogenous giant viruses drive active infection and inheritance in a multicellular host" was under peer-review at Nature Microbiology. It has now been seen by 3 referees, whose expertise and comments you will find at the end of this email. Although they find your work of some potential interest, they have raised a number of concerns that will need to be addressed before we can consider publication of the work in Nature Microbiology.

In particular, we would like Reviewer 2's point regarding whether the sense–antisense long-read mapping at the telIRL site reflects linear genomes or represents transient replication intermediates addressed either experimentally or detailed in your response to referees' document. In addition to this, Reviewer 3 raised a question regarding hemizygous integrations of EVEs which we would also like to see addressed in the revised manuscript. Aside from these, the remaining comments are quite straightforward.

Should further experimental data allow you to address these criticisms, we would be happy to look at a revised manuscript.

Please include a data availability statement as a separate section after Methods but before references, under the heading "Data Availability". This section should inform readers about the availability of the data used to support the conclusions of your study. This information includes accession codes to public repositories (data banks for protein, DNA or RNA sequences, microarray, proteomics data etc...), references to source data published alongside the paper, unique identifiers such as URLs to data repository entries, or data set DOIs, and any other statement about data availability. At a minimum, you should include the following statement: "The data that support the findings of this study are available from the corresponding author upon request", mentioning any restrictions on availability. If DOIs are provided, we also strongly encourage including these in the Reference list (authors, title, publisher (repository name), identifier, year). For more guidance on how to write this section please see: <http://www.nature.com/authors/policies/data/data-availability-statements-data-citations.pdf>

* If you have not done so already we suggest that you begin to revise your manuscript so that it conforms to our Article format instructions at <http://www.nature.com/nmicrobiol/info/final-submission>. Refer also to any guidelines provided in this letter.

* Include a revised version of any required reporting checklist. It will be available to referees (and, poten-

tially, statisticians) to aid in their evaluation if the manuscript goes back for peer review. A revised checklist is essential for re-review of the paper.

When submitting the revised version of your manuscript, please pay close attention to our <https://www.nature.com/nature-portfolio/editorial-policies/image-integrity>>Digital Image Integrity Guidelines. and to the following points below:

EXTENDED DATA FIGURES

<https://mts-nmicrobiol.nature.com/cgi-bin/main.plex?el=A4Cg5BQZ5A1BNum5J2A9ftdMHHktAgF7cQ0Ybux8V5rwZ>

Note: This url links to your confidential homepage and associated information about manuscripts you may have submitted or be reviewing for us. If you wish to forward this e-mail to co-authors, please delete this link to your homepage first.

Nature Microbiology is committed to improving transparency in authorship. As part of our efforts in this direction, we are now requesting that all authors identified as 'corresponding author' on published papers create and link their Open Researcher and Contributor Identifier (ORCID) with their account on the Manuscript Tracking System (MTS), prior to acceptance. This applies to primary research papers only. ORCID helps the scientific community achieve unambiguous attribution of all scholarly contributions. You can create and link your ORCID from the home page of the MTS by clicking on 'Modify my Springer Nature account'. For more information please visit www.springernature.com/orcid.

If you wish to submit a suitably revised manuscript we would hope to receive it within 3 months. If you cannot send it within this time, please let us know. We will be happy to consider your revision, even if a similar study has been accepted for publication at Nature Microbiology or published elsewhere (up to a maximum of 3 months).

Yours sincerely,

[redacted]

Associate Editor
Nature Microbiology

Reviewer Expertise:

Referee #1: endogenous giant viruses, EVEs
Referee #2: giant DNA viruses, virus-host dynamics
Referee #3: evolutionary biology and ecology

Reviewer Comments:

Reviewer #1 (Remarks to the Author):

This study shows that the endogenous viral elements in brown algae are latent viruses that can lead to virus production under specific environmental conditions and developmental stages. Authors elegantly demonstrate this using long read sequencing, microscopy and genetics to provide mechanistic basis of free virus production in brown algae. In other words, they establish a 'cause-effect' relationship between the integrated viruses and the produced free virus particles. I believe this is an important research towards understanding the broader consequences of giant virus endogenization in diverse eukaryotes.

I think the research is solid, and have no major concerns regarding the methods or interpretation of the results. I only have a few minor comments:

a) The authors stated "Giant virus transposons are typically of prokaryotic origin" - I think this is a sweeping statement. I agree that certain MGEs in giant viruses likely originated from prokaryotes, but the authors should be careful not to generalize that all/most GV transposons are of prokaryotic origin.

Re: Thank you for bringing this to our attention; this was indeed an oversight, and we have revised the wording to "many giant virus transposons are prokaryote-like." The study we cite shows that the majority of giant virus transposons are more closely related to prokaryotic IS elements than to eukaryotic transposons, although Mariner and PiggyBac elements were also identified. We note that this classification does not encompass other forms of mobile genetic elements, such as virophages, and we fully agree that the mobile DNA repertoire of giant viruses is highly diverse.

b) Line 529: Please provide a brief description of this method - just citing the article is not sufficient in my opinion.

c) Same for line 558 in methods. Please provide more details.

Re: We thank the reviewer for the suggestions. We have now expanded the Methods section to include a brief description of the procedures used to generate *Ectocarpus* segregating progenies and to perform CRISPR–Cas-mediated transformation. These additions provide sufficient methodological detail to ensure clarity and reproducibility.

d) Line 422-423 : Please add the parameters, specifically the score cutoff and other parameters.

Re: We thank the reviewer for this comment. We were not entirely certain which specific parameters were being referred to at this point in the manuscript. To address this concern, we have expanded the corresponding Methods section to provide additional methodological detail overall, including explicit parameter settings where applicable, in order to improve clarity and reproducibility.

e) Line 471: "previous data" - please elaborate and provide citations if necessary.

Re: Citations have been added

f) Line 509: 'hmm model ' should be 'HMM' model. this should be corrected throughout the manuscript

where needed.

Re: This has been corrected

g) There is no Data availability statement or link to the relevant data, for example the RNA-seq dataset.

Re: These issues have been addressed in the revised manuscript. A Data Availability statement has been added, including the accession numbers and links to all relevant datasets, such as the RNA-seq data, to ensure full transparency and reproducibility.

Reviewer #2 (Remarks to the Author):

Nucleocytoviruses have been shown to integrate their genomes into those of their eukaryotic hosts, giving rise to distinct genomic fragments termed Giant Endogenous Viral Elements (GEVEs). These elements have been shown to be prevalent among unicellular eukaryotes, and other recent work in green algae has shown that GEVEs represent a latent stage in the viral infection cycle. This study presents another example of a GEVE capable of establishing an active infection, however, for the first time in a model multicellular organism. Using long-read sequencing, the authors resolved multiple phaeovirus viral insertions, ranging from 288 to 407 Kbp in size, across multiples *Ectocarpus* lines. They show that specific host developmental stages and environmental cues lead to the onset of viral symptoms and subsequent viral particle production. Genotyping of the haploid symptomatic and asymptomatic progeny from different lines revealed specific viral insertions that were putatively active and later shown to be transcriptionally reactivated under cold treatment. Active elements do not seem to interact with or change the severity of their associated symptoms when present alongside other inactive elements. Finally, analysis of the genomic content of these GEVEs, together with characterization of their end termini, supports a model in which viral genome integration and excision occur via a tyrosine recombinase interacting with attB/attP-like sites, paired with genome linearization mediated by a telomerase resolvase.

Overall, this is a well-written and meticulously performed study that highlights an exciting system to explore latency in nucleocytoviruses of multicellular organisms. Although, phaeovirus signatures have been shown to be prevalent among brown algal genomes, the role of latency in their infection cycles has remained unclear. This study provides strong evidence for new integration and reactivation dynamics of GEVEs and highlights the importance of host biology and environmental context in modulating the reactivation of integrated phaeoviruses in natural settings.

One unresolved question in the manuscript that was highlighted in the discussion, is whether the sense–antisense long-read mapping at the telRL site reflects linear genomes with covalently closed telomeres or instead represents transient replication intermediates of the viral genome. Long-read sequencing of the purified viral particles DNA could help resolve if the linear form is packaged in mature virions. An assessment of the proportion of reads spanning the attP site and those showing sense–antisense mapping at the telRL site, ideally in a time-series during induction, could help resolve the sequence of events during viral reactivation. We understand if these experiments are outside of the scope of this manuscript.

Re: As of now, we are unable to distinguish between a linear viral genome with covalently closed telomeres and transient replication intermediates forming head-to-head and tail-to-tail concatemers. From the current Nanopore sequencing reads, the ratio of reads mapping sense–antisense to those spanning the telRL site is roughly 1.7 (coverage: telLL' site = 2100; telRR' site = 2300; telRL site = 1300; overall coverage of the EVE \approx 7000, overall coverage of the host \approx 100).

The final form of the viral genome packaged into virions is indeed an important point. Although we do not currently have a protocol to purify sufficient DNA from viral-like particles for long-read sequencing, we were able to probe the virion DNA by PCR. Both the attP and telRL sites could be amplified, suggesting that the viral DNA is packaged, at least in part, in a circular form. We cannot, however, exclude the possibility that multiple forms of viral DNA coexist in the sample. Contamination from algal DNA can be excluded, as neither the attR site nor the algal ITS1 region (from the rRNA array) could be amplified. These results are presented in Figure 3F and discussed in the main text (lines 299–302).

We propose the following model: the EVEc excises from the genome, becomes linearized, and replicates as a linear molecule, before being circularized again prior to packaging into virions.

Regarding time-resolved analyses, the asynchronous nature of gametangia production and the presence of mixed cell populations at different stages of viral infection make such experiments practically impossible in *Ectocarpus*. While time-series data could theoretically help clarify replication dynamics, the biological constraints of the system render this approach unfeasible, and we therefore consider it beyond the scope of the current manuscript.

Minor comments

In the abstract I would shy away from statements of priority (“the first direct evidence” could just be “direct evidence”) – some journals advise against these statements given the nuance often involved in these statements, and I don't think it is needed here given the clear advances presented. I do not think that mitigates the importance of this work at all.

Re: we have deleted the word ‘first’

The text on introns is intriguing. Spliceosomal introns have also been found in chloroviruses, and it would be worth citing that here for context – Fitzgerald et al Jvi doi: 10.1016/j.virol.2006.08.033 and Sun et al Journal of Molecular Evolution <https://doi.org/10.1007/s002399910009>

Re: Thank you for pointing these interesting articles. Citation have been added.

P.2 Ln.51: The FirrV-1 genome has been partially sequenced. I would suggest indicating that only these two genomes have been sequenced in their entirety.

Re: The sentence has be modified as follows: , “Only two phaeovirus genomes have been sequenced in their entirety, *Ectocarpus siliculosus* virus 1 (EsV-1)¹⁸ and *Feldmannia* species virus 158 (FsV-158)¹⁹”

P.4 Ln.129: In consistency with Fig. S6 and Fig. S7 termini names, should the “CC” dinucleotide form part of the right terminus instead?

Re: We modified the figures for consistency and chose to present each C as part of one termini.

P.6 Ln.197: Text indicate that 110 haploid individuals were obtained for line Ec267, but Fig. 2A shows n=140.

We thank the reviewer for noticing this discrepancy. The correct number is 140 haploid individuals for line Ec267, as shown in Figure 2A. The text has been corrected accordingly.

P.7 Ln. 218: To help in the interpretation of Fig. 2G, I would suggest clarifying briefly what constitute as viral symptoms according to the microscopy results. Indicate in the caption of Fig. 2G that these were observed in the gametangia.

Re: we thank you for the suggestion, this has been done.

P.8. Ln.264,266: Figure 3C refers to the TEM and not the telIRL site.

Re: This has been corrected

P.9 Ln. 279: Should it be Fig. 2B,G?

Re: Corrected

P.28 Fig.S2: In A) EVEb is missing, and in B) EVEc is misspelled.

Re: Corrected (EVEb was present but not highlighted in black)

P.29 Fig.S3: Keep consistency in the caption and figure, either ITR or TIR.

Re: done

Typos and changes for consistency:

P.1 Ln.17: do not capitalize phaeoviruses if referred to as a collective group of viruses. Check for other

instance throughout the text.

P.2 Ln.41: Italicize Nucleocytoviricota. Check for other instance throughout the text.

Ln.163: do not capitalize xanthoviruses if referred to as a collective group of viruses. Check for other instance throughout the text.

P.10 Ln.332: do not capitalize punuiviruses if referred to as a collective group of viruses. Check for other instance throughout the text.

P.13 Ln. 414: add space in "100µm".

P.13 Ln. 417: change "grinded" to "ground".

P.13 Ln. 420: change "RNaseA" to "RNase A".

P.13 Ln. 421: "µLProteinaseK" to "µL Proteinase K". Indicate Proteinase K concentration.

P.13 Ln. 422: add space in "200µL". Change "RNaseA" to "RNase A".

P.13 Ln. 423: add space in "1.5h".

P.13 Ln. 427: change "Qbit" to "Qubit".

P.14 Ln. 460: change "RNA-seq" to "RNA-Seq".

P.14 Ln. 469: change "trimal" to "trimAl".

P.14 Ln. 470: change "IQtree" to "IQ-TREE".

P.14 Ln.472: add space in "DNAPolymerase".

P.14 Ln.478: change "Tree" to "The tree".

P.15 Ln.519: change "cd-hit" to "CD-HIT".

P.15 Ln. 521: change "clustalo, trimmed with trimal" to "ClustalO, and trimmed with trimAl".

P.16 Ln. 534: change "Material" to "The material".

P.16 Ln.537-538: change "grinded" to "ground". Change "for two times" to "twice for".

P.16 Ln.541: change "grinded" to "ground".

P.16 Ln.545: change "40 cycles" to "40 cycles of amplification".

P.16 Ln.550: change "PCR were performed on the same grinded algal" to "PCR was performed on the same ground algal".

P.16 Ln.552: I suggest enclosing "see Table S7 for the amplification cycle programs" in parenthesis.

P.16 Ln.557: change "See" to "see".

P.17 Ln.585: change "DNAse" to "DNase".

P.17 Ln.588: missing a period.

P.27 Ln.795-796: misspelled "meiosis" and "Meiospores".

Re: Thank you for pointing out these typos, which have been corrected.

Reviewer #3 (Remarks to the Author):

Duchene et al. present a comprehensive study demonstrating the presence of an active giant provirus in a multicellular host. This is quite groundbreaking and exciting, as this type of viral lifestyle has so far been attributed only to smaller viruses and mostly in unicellular hosts. The findings, thus, open a new window into viral and microbial ecology with far-reaching implications.

I found the paper to be well-written and clear. The different aspects of the findings are analyzed comprehensively, using complementary approaches and angles, and in my opinion, provide solid support for the reported findings. This includes thorough genomic analysis, a convincing hypothesis for the integration mechanism, and a compelling demonstration of temperature-induced activation.

I only have a few minor comments/questions

Diploid assemblies with hemizygous integrations can be tricky. In particular, exiting hemizygous integration in one locus can be missed if the same integration also already exists at a different locus. Such an effect could be checked by investigating the depth of read coverage between EVEs and host contigs. In

case of a single, hemizygous integration, EVE coverage should be half that of the mean host coverage. Higher coverage of the EVE could indicate the presence of additional, unobserved integrations

Re: As suggested, we examined read coverage of EVEs relative to host chromosomes to assess the presence of hemizygous integrations. The results are shown in the figure below.

For the diploid strain Ec17, coverage of each EVE is approximately half (or slightly above half) that of the host genome, confirming that each EVE is present in a single copy. The diploid strain Ec267 contains 4 times the same EVE, and indeed the coverage of the EVE is roughly twice that of the algal genome. These observations give us confidence that all viral integrations have been correctly identified in these genomes. While we are not convinced that including this plot in the current version of the manuscript would substantially strengthen the presentation, we would be happy to add it if the editor and reviewer consider it important.

According to the methods, the authors grew the hosts for a relatively long time under different temperature conditions (4 weeks at 10°C) before analyzing viral activity via RNA seq. I was wondering if it also takes that long to observe symptoms after giving the trigger, or if it is also possible to trigger viral production in much shorter periods of coldness, and if it would be even possible to detect exactly when activation starts?

Re: We thank the reviewer for this question. The timing of viral activation relative to the cold trigger is not precisely known, as we do not yet know the exact stage during gametophyte development at which induction occurs. Gametophyte development from meiospores to maturity takes approximately seven weeks, and in our protocol, we allow gametophytes to reach maturity at 10 °C to ensure consistent and reliable induction. To date, no alternative cultivation method or experimental setup has been established that allows the onset of viral symptoms to be observed reliably in shorter time frames or at specific developmental stages. Therefore, we follow the classical long-term protocol to maximize reproducibility and the likelihood of detecting viral activity by RNA-seq.

There might be a typo in the M&M: MCsan > MCscan?

Re: this has been corrected